# MULTI-SCALE HYPERGRAPH MEETS LLMS: ALIGNING LARGE LANGUAGE MODELS FOR TIME SERIES ANALYSIS

**Zongjiang Shang**[1,2], **Dongliang Cui**[1,2], **Binqing Wu**[1,2], **Ling Chen** [1,2*]
[1] State Key Laboratory of Blockchain and Data Security, Zhejiang University
[2] College of Computer Science and Technology, Zhejiang University
{zongjiangshang, runnercdl, binqingwu, lingchen}@cs.zju.edu.cn

## ABSTRACT

Recently, there has been great success in leveraging pre-trained large language models (LLMs) for time series analysis. The core idea lies in effectively aligning the modality between natural language and time series. However, the multi-scale structures of natural language and time series have not been fully considered, resulting in insufficient utilization of LLMs capabilities. To this end, we propose MSH-LLM, a Multi-Scale Hypergraph method that aligns Large Language Models for time series analysis. Specifically, a hyperedging mechanism is designed to enhance the multi-scale semantic information of time series semantic space. Then, a cross-modality alignment (CMA) module is introduced to align the modality between natural language and time series at different scales. In addition, a mixture of prompts (MoP) mechanism is introduced to provide contextual information and enhance the ability of LLMs to understand the multi-scale temporal patterns of time series. Experimental results on 27 real-world datasets across 5 different applications demonstrate that MSH-LLM achieves the state-of-the-art results.

## 1 INTRODUCTION

Time series analysis is a critical ingredient in a myriad of real-world applications, e.g., forecasting (Liu et al., 2023b; Wan et al., 2024; Shang et al., 2024a), imputation (Wang et al., 2024a), and classification (Chen et al., 2024b; Wang et al., 2024c), which is applied across diverse domains, including retail, transportation, economics, meteorology, healthcare, etc. In these real-world applications, the task-specific models usually require domain knowledge and custom designs (Chen et al., 2024a; Zhou et al., 2023a). This contrasts with the demand of time series foundation models, which are designed to perform well in diverse applications, including few-shot learning and zero-shot learning, where minimal and no training data is provided.

Recently, pre-trained foundation models, especially large language models (LLMs), have achieved great success across many fields, e.g., natural language processing (NLP) (Touvron et al., 2023; Achiam et al., 2023; Radford et al., 2021) and computer vision (CV) (Wang et al., 2024b; Pi et al., 2024). Although the lack of large pre-training datasets and a consensus unsupervised objective makes it difficult to train foundation models for time series analysis from scratch (Sun et al., 2024; Jin et al., 2024; Pan et al., 2024), the fundamental commonalities between natural language and time series in sequential structure and contextual dependency provide an avenue to apply LLMs for time series analysis. The core idea lies in the effective alignment of the modality between natural language and time series, either by reprogramming the input time series (Xue & Salim, 2023; Cao et al., 2024) or by introducing prompts to provide contextual information for the input time series (Sun et al., 2024; Kamarthi & Prakash, 2023; Jin et al., 2024).

In the process of aligning LLMs for time series analysis, we observe that both natural language and time series present multi-scale structures. In natural language, multi-scale structures typically manifest as semantic structures at different scales (Yang et al., 2024b), e.g., words, phrases, and

---

*Corresponding author.

sentences. In time series, the multi-scale structures often demonstrate as multi-scale temporal patterns (Wen et al., 2021; Liu et al., 2021; Shang et al., 2024a). For example, influenced by periodic human activities, traffic volume and energy usage exhibit pronounced daily patterns (e.g., afternoon or evening) and weekly patterns (e.g., weekday or weekend). Considering multi-scale alignment between natural language and time series enables models to learn richer representations and enhance their cross-modality learning abilities. However, we argue that performing multi-scale alignment is a non-trivial task, as two notable problems need to be addressed.

The first problem lies in the disparity between the multi-scale semantic space of natural language and that of time series. The multi-scale semantic space of natural language is both distinctive and informative (Pan et al., 2024), while the multi-scale semantic space of time series faces the semantic information sparsity problem due to an individual time point containing less semantic information (Shang et al., 2024b; Chang et al., 2024). This disparity makes it difficult to leverage off-the-shelf LLMs for time series analysis. To tackle this, most existing works employ patch-based methods (Nie et al., 2022; Jin et al., 2024) to capture group-wise interactions and enhance the semantic information of time series semantic space. However, simple partitioning of patches may introduce noise interference and make it hard to discover implicit interactions.

The second problem when performing multi-scale alignment lies in the knowledge and reasoning capabilities to interpret temporal patterns are not naturally present within the pre-trained LLMs. To unlock the knowledge within LLMs and activate their reasoning capabilities for time series analysis, existing methods introduce prefix prompts (Jin et al., 2024; Liu et al., 2024) or self-prompt mechanisms (Sun et al., 2024) to provide task instruction and enrich the input contextual information. While these methods are intuitive and straightforward, they struggle to understand temporal patterns due to their failure to leverage multi-scale temporal features. Therefore, it is still an open challenge to design prompts that are accurate, data-correlated, and task-specific.

To this end, we propose MSH-LLM, a **M**ulti-**S**cale **H**ypergraph method that aligns **L**arge **L**anguage **M**odels for time series analysis. To the best of our knowledge, MSH-LLM is the first multi-scale alignment work for time series analysis, which leverages the hyperedging mechanism to enhance the multi-scale semantic information of time series and employs the mixture of prompts mechanism to enhance the ability of LLMs in understanding multi-scale temporal patterns. Our contributions are given as follows:

- We develop a hyperedging mechanism that leverages learnable hyperedges to extract hyperedge features with group-wise information from multi-scale temporal features, which can enhance the multi-scale semantic information of time series semantic space while reducing irrelevant information interference.
- We propose a cross-modality alignment module to perform multi-scale alignment based on the multi-scale prototypes and hyperedge features, which goes beyond relying solely on single-scale alignment and obtains richer representations. In addition, we design a mixture of prompts (MoP) mechanism, which augments the input contextual information with different prompts to enhance the reasoning ability of LLMs for time series analysis.
- We evaluate MSH-LLM on 27 real-world datasets across 5 different applications. The experimental results demonstrate that MSH-LLM consistently outperforms existing methods, highlighting its effectiveness in activating the capability of LLMs for time series analysis.

## 2 RELATED WORK

**In-Modality Learning Methods.** Recent studies in NLP (Devlin, 2018; Radford et al., 2019; Brown, 2020; Touvron et al., 2023) and CV (Touvron et al., 2021; Wang et al., 2023; Bao et al., 2022) have shown that pre-trained foundation models can be fine-tuned for various downstream tasks within the same modality, significantly reducing the need for costly training from scratch while maintaining high performance. BERT (Devlin, 2018) uses bidirectional encoder representations from transformers to recover the randomly masked tokens of the sentences. GPT3 (Brown, 2020) trains a transformer decoder on a large language corpus with much more parameters, which can be utilized for diverse applications. BEiT (Bao et al., 2022) designs a masked image modeling task to pretrain vision transformers. Motivated by the above, recent time series pre-trained models use different strategies, e.g., supervised learning methods (Fawaz et al., 2018) or self-supervised learning methods (Chen

et al., 2025; Woo et al., 2022a), to learn representations across diverse domains and then fine-tune on similar applications to perform specific tasks. However, due to the lack of large pre-training datasets and a consensus unsupervised objective, it is difficult to train foundation models for general-purpose time series analysis that covers diverse applications.

**Cross-Modality Learning Methods.** Due to the fundamental commonalities between natural language and time series in sequential structure and contextual dependency, recent works have explored cross-modality learning by applying LLMs for time series analysis (Bian et al., 2024; Zhou et al., 2023a; Liu et al., 2024; Jin et al., 2024). FPT (Zhou et al., 2023a) represents an early attempt to fine-tune key parameters of LLMs and adapt them for time series analysis. aLLM4TS (Bian et al., 2024) introduces a two-stage pre-training strategy that first performs causal next-patch training and then enacts a fine-tuning strategy for downstream tasks. However, fine-tuning LLMs for training and inference can sometimes be resource-consuming due to the immense size of LLMs (Liu et al., 2024). Some recent works have explored the alignment of frozen LLMs for time series analysis, either by reprogramming the input time series or introducing prompts to provide contextual information for the input time series. Time-LLM (Jin et al., 2024) proposes a reprogramming strategy that aligns time series inputs with textual prototypes before passing them to a frozen LLM. AutoTimes (Liu et al., 2024) repurposes frozen LLMs as autoregressive time series forecasters and introduces relevant time series prompts to enhance forecasting. Although these methods achieve promising results, they overlook the multi-scale structures of natural language and time series.

**Multi-Scale Time Series Analysis Methods.** Existing multi-scale time series analysis methods are aimed at modeling temporal pattern interactions at different scales (Chen et al., 2021; Shang et al., 2024b; Chen et al., 2023b). TAMS-RNNs (Chen et al., 2021) disentangles input series into multi-scale representations and uses different update frequencies to model multi-scale temporal pattern interactions. Benefiting from the attention mechanism, transformers achieve promising results in time series analysis. Pyraformer (Liu et al., 2021) treats multi-scale features as nodes and leverages pyramidal attention to model interactions between nodes at different scales. To solve the problem of semantic information sparsity, Pathformer (Chen et al., 2023b) divides time series into multiple resolutions using patches of different sizes and uses the dual attention to capture group-wise pattern interactions at different scales. MSHyper (Shang et al., 2024b) combines Transformer with multi-scale hypergraphs to capture group-wise pattern interactions across different temporal scales. However, fixed segments or pre-defined rules cannot capture implicit pattern interactions and may introduce noise interference.

In this paper, we find that both natural language and time series present multi-scale structures. Therefore, we propose a multi-scale hypergraph method that aligns large language models (LLMs) for time series analysis. Specifically, a hyperedging mechanism is introduced to enhance the multi-scale semantic information of time series semantic space and reduce noise interference. Then, a cross-modality alignment (CMA) module is introduced to perform multi-scale alignment. In addition, we develop a mixture-of-prompts (MoP) strategy to strengthen the reasoning ability of LLMs over multi-scale temporal patterns.

## 3 PRELIMINARIES

**Hypergraph.** A hypergraph can be represented as $\mathcal{G} = \{\mathcal{V}, \mathcal{E}\}$, where $\mathcal{V} = \{v_i\}_{i=1}^N$ represents the set of nodes and $\mathcal{E} = \{e_j\}_{j=1}^M$ denotes the set of hyperedges. Unlike simple graphs, each hyperedge $e \in \mathcal{E}$ characterizes group-wise interactions by encompassing an arbitrary subset of nodes. The structural topology can represented by the incidence matrix $\mathbf{H} \in \mathbb{R}^{N \times M}$, where $\mathbf{H}_{nm} = 1$ signifies the membership of node $v_n$ in hyperedge $e_m$, and $\mathbf{H}_{nm} = 0$ otherwise. More descriptions of hypergraph learning are provided in Appendix B.

**Problem Definition.** The proposed MSH-LLM is designed to align frozen LLMs for time series analysis, which covers different applications across various domains. For a given specific application that consists the input time series $\mathbf{X}^{\mathrm{I}}_{1:T} \in \mathbb{R}^{T \times D}$ with $T$ time steps and $D$ dimensions, the goal of time series analysis is to predict important properties of the time series. For example, the forecasting task aims at predicting the future $H$ steps $\mathbf{X}^{\mathrm{O}}_{T+1:T+H} \in \mathbb{R}^{H \times D}$, while the classification task aims at predicting the class labels of the given time series.

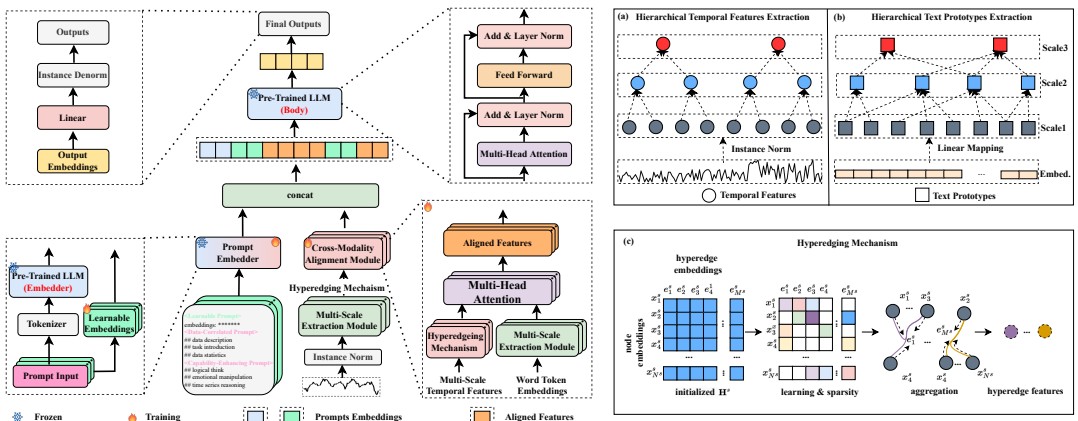

Figure 1: The framework of MSH-LLM. (a) and (b) provide detailed delineation of the multi-scale extraction module, while (c) elaborates on the hyperedging mechanism.

## 4 METHODOLOGY

MSH-LLM aims to repurpose frozen large language models (LLMs), such as LLaMA (Touvron et al., 2023) and GPT-2 (Radford et al., 2019), for general-purpose time series analysis. This is achieved by considering the multi-scale structures inherent in both natural language and time series data. As depicted in Figure 1, the time series data and word token embeddings (derived from pre-trained LLMs) are mapped into multi-scale temporal features and text prototypes, respectively. Then, a hyperedging mechanism is developed to enhance the multi-scale semantic information of time series semantic space, and a cross-modality alignment (CMA) module is introduced to align the modality between natural language and time series. In addition, a mixture of prompts (MoP) mechanism is introduced to provide multi-scale contextual information and enhance the ability of LLMs in understanding multi-scale temporal patterns of time series.

### 4.1 MULTI-SCALE EXTRACTION (ME) MODULE

The ME module is designed to extract the multi-scale features, which include hierarchical temporal features extraction and hierarchical text prototypes extraction.

**Hierarchical Temporal Features Extraction.** As illustrated in Figure 1(a), given input time series $\mathbf{X}^1 = \mathbf{X}^{\mathrm{I}}_{1:T}$, we first normalize it through reversible instance normalization (Kim et al., 2021). Then, we perform hierarchical temporal features extraction, which can be formulated as follows:

$$\mathbf{X}^s = Agg(\mathbf{X}^{s-1}; \theta^{s-1}) \in \mathbb{R}^{N^s \times D}, s \geq 2, \tag{1}$$

where $\mathbf{X}^s = \{\boldsymbol{x}^s_t | \boldsymbol{x}^s_t \in \mathbb{R}^D, t \in [1, N^s]\}$ denotes the feature set at scale $s$, $s = 2, ..., S$ denotes the scale index, and $S$ is the total number of scales. $Agg$ is the aggregation function, such as 1D convolution or average pooling. $\theta^{s-1}$ denotes the learnable parameters of the aggregation function at scale $s - 1$, $N^s = \left\lfloor \frac{N^{s-1}}{l^{s-1}} \right\rfloor$ is the sequence length at scale $s$, and $l^{s-1}$ denotes the size of the aggregation window at scale $s - 1$.

**Hierarchical Text Prototypes Extraction.** The hierarchical text prototypes extraction aims to map word token embeddings in natural language to multi-scale structures, e.g., words, phrases, and sentences, for alignment with multi-scale temporal features. As shown in Figure 1(b), given the word token embeddings based on pre-trained LLMs $\mathbf{U} \in \mathbb{R}^{V \times P}$, where $V$ is the vocabulary size and $P$ is the hidden dimension of LLMs. We first transform them to a small collection of text prototypes through linear mapping, which can be represented as $\mathbf{U}^1 \in \mathbb{R}^{V' \times P}$, where $V' \ll V$. This approach is efficient and can capture key linguistic signals related to time series. Then, we can derive multi-scale text prototypes via linear mapping, as formulated below:

$$\mathbf{U}^s = Linear(\mathbf{U}^{s-1}; \lambda^{s-1}) \in \mathbb{R}^{V^s \times P}, s \geq 2, \tag{2}$$

where $Linear$ denotes the linear mapping function, $\mathbf{U}^s$ denotes the text prototypes at scale $s$, while $\lambda^{s-1}$ refers to the learnable parameters of the linear mapping function at scale $s-1$. After mapping, we aim for the multi-scale text prototypes to capture the linguistic signals that describe multi-scale temporal patterns. Experimental results in Appendix G validate the effectiveness of the multi-scale text prototype extraction compared to manually selected approaches.

## 4.2 HYPEREDGING MECHANISM

After obtaining the multi-scale temporal features and text prototypes, a straightforward way to align LLMs for time series analysis is to perform cross-modality alignment at different scales. However, the semantic space disparity poses a significant challenge, making it difficult to leverage the off-the-shelf LLMs for time series analysis. To tackle this, some recent studies (Jin et al., 2024; Shang et al., 2024a) show that group-wise interactions can help enrich the semantic information of time series semantic space, thereby enhancing its consistency with the semantic space of natural language. Therefore, we introduce a hyperedging mechanism that utilizes learnable hyperedges to capture group-wise interactions at different scales.

As depicted in Figure 1(c), we first treat multi-scale temporal features as nodes and initialize two kinds of learnable embeddings at scale $s$, i.e., hyperedge embeddings $\boldsymbol{E}_{\text{hyper}}^s \in \mathbb{R}^{M^s \times D}$ and node embeddings $\boldsymbol{E}_{\text{node}}^s \in \mathbb{R}^{N^s \times D}$, where $M^s$ is a hyperparameter that defines the number of hyperedges at scale $s$. Then, the similarity calculation is performed to construct the scale-specific incidence matrix $\mathbf{H}^s$, which can be formulated as follows:

$$\begin{aligned} \boldsymbol{U}_1^s &= tanh(\boldsymbol{E}_{\text{nodes}}^s \boldsymbol{\beta}), \\ \boldsymbol{U}_2^s &= tanh(\boldsymbol{E}_{\text{hyper}}^s \boldsymbol{\varphi}), \\ \mathbf{H}^s &= Linear(ReLU(\boldsymbol{U}_1^s (\boldsymbol{U}_2^s)^T)), \end{aligned} \tag{3}$$

where $\boldsymbol{\beta} \in \mathbb{R}^{1 \times 1}$ and $\boldsymbol{\varphi} \in \mathbb{R}^{1 \times 1}$ are learnable parameters. The $tanh$ activation function is used to perform nonlinear transformations and the $ReLU$ activation function is applied to eliminate weak connections. To enhance the robustness of the model, reduce the computation cost of subsequent operations, and mitigate the impact of noise, we introduce a sparsity strategy to make $\mathbf{H}^s$ sparse, which can be formulated as follows:

$$\mathbf{H}_{nm}^s = \begin{cases} 1, & \mathbf{H}_{nm}^s \in TopK(\mathbf{H}_{n*}^s, \eta) \\ 0, & \mathbf{H}_{nm}^s \notin TopK(\mathbf{H}_{n*}^s, \eta) \end{cases} \tag{4}$$

where $\eta$ is the threshold of $TopK$ function and represents the maximum number of neighboring hyperedges associated with a node.

The final scale-specific incidence matrices can be represented as $\{\mathbf{H}^1, \cdots, \mathbf{H}^s, \cdots, \mathbf{H}^S\}$ and the hyperedge features of the $i$th hyperedge $e_i^s \in \boldsymbol{\mathcal{E}}^s$ based on the scale-specific incidence matrix at scale $s$ is formulated as follows:

$$e_i^s = Avg\left(\sum_{x_j^s \in \mathcal{N}(e_i^s)} \boldsymbol{x}_j^s\right) \in \mathbb{R}^D, \tag{5}$$

where $Avg$ is the average operation, $\mathcal{N}(e_i^s)$ is the neighboring nodes connected by $e_i^s$ at scale $s$, and $\boldsymbol{x}_j^s \in \mathbf{X}^s$ represents the $j$th node features at scale $s$. The final hyperedge feature set at different scales can be represented as $\{\boldsymbol{\mathcal{E}}^1, \cdots, \boldsymbol{\mathcal{E}}^s, \cdots, \boldsymbol{\mathcal{E}}^S\}$.

Compared with other methods, our hyperedging mechanism is novel in two aspects. Firstly, our methods can capture implicit group-wise interactions at different scales in a learnable manner, while most existing methods (Nie et al., 2022; Zhou et al., 2023a; Shang et al., 2024b) rely on pre-defined rules to model group-wise interactions at a single scale. Secondly, although some methods (Shang et al., 2024a; Jiang et al., 2019) learn from hypergraphs, they focus on constraints or clustering-based approaches to learn the hypergraph structures. In contrast, MSH-LLM learns hypergraph structures in a data-driven way by incorporating learnable parameters and nonlinear transformations, which is more flexible and can learn more complex hypergraph structures.

## 4.3 CROSS-MODALITY ALIGNMENT (CMA) MODULE

The CMA module is designed to align the modality between natural language and time series based on the multi-scale hyperedge features and text prototypes. To achieve this, a multi-head cross-attention

is used to perform alignment at different scales. Specifically, for the given text prototpyes $\mathbf{U}^s$ and hyperedge features $\boldsymbol{\mathcal{E}}^s$ at scale $s$, we first transform it into query $\mathbf{Q}_{\jmath}^s = \boldsymbol{\mathcal{E}}^s \mathbf{W}_{\mathrm{q},\jmath}^s$, key $\mathbf{K}_{\jmath}^s = \mathbf{U}^s \mathbf{W}_{\mathrm{k},\jmath}^s$, and value $\mathbf{V}_{\jmath}^s = \mathbf{U}^s \mathbf{W}_{\mathrm{v},\jmath}^s$, respectively, where $\jmath = 1, ..., \mathcal{J}$ denotes the head index. $\mathbf{W}_{\mathrm{q},\jmath}^s \in \mathbb{R}^{D \times d}$, $\mathbf{W}_{\mathrm{k},\jmath}^s \in \mathbb{R}^{P \times d}$, and $\mathbf{W}_{\mathrm{v},\jmath}^s \in \mathbb{R}^{P \times d}$ are learnable weight matrices at scale $s$, $d = \lfloor \frac{D}{\mathcal{J}} \rfloor$. Then, the multi-head cross-attention is applied to align the hyperedging features with text prototypes, which can be formulated as follows:

$$\boldsymbol{\mathcal{Z}}_{\jmath}^s = Attn(\mathbf{Q}_{\jmath}^s, \mathbf{K}_{\jmath}^s, \mathbf{V}_{\jmath}^s) = softmax(\frac{\mathbf{Q}_{\jmath}^s (\mathbf{K}_{\jmath}^s)^{\top}}{\sqrt{d}}) \mathbf{V}_{\jmath}^s. \tag{6}$$

Then, we aggregate $\boldsymbol{\mathcal{Z}}_k^s$ in every head to obtain the output of multi-head attention $\boldsymbol{\mathcal{Z}}^s \in \mathbb{R}^{M^s \times D}$ at scale $s$. The final aligned features at different scales can be represented as $\{\boldsymbol{\mathcal{Z}}^1, \cdots, \boldsymbol{\mathcal{Z}}^s, \cdots, \boldsymbol{\mathcal{Z}}^S\}$.

## 4.4 MIXTURE OF PROMPTS (MoP) MECHANISM

The performance of LLMs depends significantly on the design of the prompts used to steer the model capabilities (Pan et al., 2024; Zhou et al., 2023b). To enhance the reasoning capabilities of LLMs, most existing methods focus on prefix prompts (Jin et al., 2024; Liu et al., 2024) or self-prompt mechanisms (Sun et al., 2024; Lester et al., 2021) to provide task instructions and enrich the input contextual information. However, the prompts affecting the reasoning capabilities of LLMs are multifaceted. Relying on a single type of prompt cannot fully activate the reasoning capabilities of LLMs. Therefore, we propose a MoP mechanism, which augments the input contextual information with different prompts (i.e., learnable prompts, data-correlated prompts, and capability-enhancing prompts) and enhances the reasoning capabilities of LLMs towards multi-scale temporal patterns.

**Learnable Prompts.** Learnable or soft prompts show great effectiveness across many fields by utilizing learnable embeddings, which are learned from the supervised loss between the output of the model and the ground truth. However, existing learnable prompts cannot capture the temporal dynamics from multi-scale temporal patterns.

Therefore, we introduce multi-scale learnable prompts $\boldsymbol{\mathcal{C}}_l = \{\mathbf{P}^1, ..., \mathbf{P}^s, ..., \mathbf{P}^S\}$, where $\mathbf{P}^s \in \mathbb{R}^{L^s \times D}$ is the scale-specific prompts and $L^s$ is the prompt length at scale $s$. $\boldsymbol{\mathcal{C}}_l$ learns from the loss between the output of LLMs and task-specific ground truth.

**Data-Correlated Prompts.** As shown in Figure 2(a), we introduce three components to construct data-correlated prompts $\boldsymbol{\mathcal{C}}_d$, i.e., data description ($\pi$), task introduction ($\tau$), and data statistics ($\mu$). The data description provides LLMs with essential background information about the input time series, the task introduction is used to guide LLMs in understanding and performing specific tasks, and the data statistics provide time series statistics that include both input sequence and sub-sequences at different scales. The final data-correlated prompts can be formulated as follows:

$$\boldsymbol{\mathcal{C}}_d = LLMs(tokenizer(\pi, \tau, \mu)). \tag{7}$$

**Capability-Enhancing Prompts.** Some recent studies in NLP (Kojima et al., 2022) and CV (Ge et al., 2023) have shown that prompt engineering, e.g., template and chain-of-thought prompts, can significantly enhance the reasoning abilities of LLMs, especially for few-shot or zero-shot learning. We have observed similar rules when aligning LLMs for time series analysis. Therefore, as shown in Figure 2(b), we design three components to construct capability-enhancing prompts $\boldsymbol{\mathcal{C}}_c$, i.e., logical thinking ($\phi$), emotional

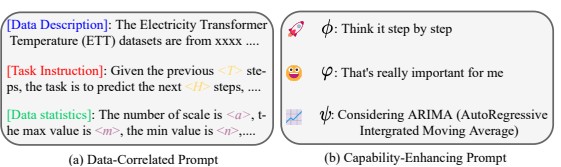

(a) Data-Correlated Prompt    (b) Capability-Enhancing Prompt

Figure 2: Prompt example. $<>$ and $<>$ are task-specific configurations and input statistic information, respectively.

manipulation ($\varphi$), and time series reasoning correlated prompts ($\psi$). The logical thinking prompts guide LLMs to solve problems in a step-by-step manner, which may enhance the multi-step reasoning abilities of LLMs; The emotional manipulation prompts mimic the impact of emotions on human decision-making, using "emotional blackmail" to make the model focus more on the current task; The time series reasoning correlated prompts provide specific methodologies that help LLMs to deal with temporal features. The final capability-enhancing prompts are formulated as follows:

$$\boldsymbol{\mathcal{C}}_c = LLMs(tokenizer(\phi, \varphi, \psi)). \tag{8}$$

## 4.5 OUTPUT PROJECTION

After obtaining the MoP, we first concatenate the learnable prompts with the aligned features at different scales, then concatenate them with data-correlated prompts and capability-enhancing prompts and put them into LLMs to get the output representations, which can be formulated as follows:

$$\mathcal{O} = LLMs([\mathcal{C}_d, \mathcal{C}_c, [\mathbf{P}^1, \mathcal{Z}^1], ..., [\mathbf{P}^S, \mathcal{Z}^S]]). \tag{9}$$

where $[.,.]$ denotes the concatenation operation. Then, we obtain the final results through linear mapping and instance denormalization.

## 5 EXPERIMENTS

**Experimental Settings.** We conduct experiments on 27 real-world datasets across 5 different applications to verify the effectiveness of MSH-LLM, including long/short-term time series forecasting, classification, few-shot learning, and zero-shot learning. Overall, MSH-LLM achieves state-of-the-art results in a range of critical time series analysis tasks against 19 advanced baselines. More details about baselines, datasets, and experiment settings are given in Appendix A, C, and D, respectively.

### 5.1 LONG-TERM FORECASTING

**Setups.** For long-term time series forecasting, we evaluate the performance of MSH-LLM on 7 commonly used datasets, including ETT (i.e., ETTh1, ETTh2, ETTm1, and ETTm2), Weather, Traffic, and Electricity datasets. More details about the datasets are given in Appendix C. Following existing works (Jin et al., 2024; Zhou et al., 2023a; Pan et al., 2024), we set the input length $T = 512$ and the forecasting lengths $H \in \{96, 192, 336, 720\}$. The mean square error (MSE) and mean absolute error (MAE) are set as the evaluation metrics.

Table 1: Long-term time series forecasting results. Results are averaged from all forecasting lengths. Smaller values correspond to better performance. **Bolded**: Best, Underlined: Second best. Full results are listed in Appendix F.1, Table 8.

| Methods | MSH-LLM (Ours) | | S$^2$IP-LLM (ICML 2024) | | Time-LLM (ICLR 2024) | | AutoTimes (NeurIPS 2024) | | FPT (NeurIPS 2023) | | AMD (AAAI 2025) | | ASHyper (NeurIPS 2024) | | iTransformer (ICLR 2024) | | MSHyper (arXiv 2024) | | DLinear (AAAI 2023) | | TimesNet (ICLR 2023) | | FEDformer (ICML 2022) | |
|---|---|---|---|---|---|---|---|---|---|---|---|---|---|---|---|---|---|---|---|---|---|---|---|---|
| Metirc | MSE | MAE | MSE | MAE | MSE | MAE | MSE | MAE | MSE | MAE | MSE | MAE | MSE | MAE | MSE | MAE | MSE | MAE | MSE | MAE | MSE | MAE | MSE | MAE |
| Weather | **0.217** **0.254** | | 0.223 0.259 | | 0.231 0.269 | | 0.233 0.279 | | 0.237 0.271 | | 0.225 0.265 | | 0.254 0.283 | | 0.305 0.335 | | 0.243 0.271 | | 0.249 0.300 | | 0.259 0.287 | | 0.309 0.360 | |
| Electricity | **0.159** **0.253** | | 0.163 0.258 | | 0.165 0.261 | | 0.162 0.261 | | 0.167 0.263 | | 0.162 0.257 | | 0.162 **0.253** | | 0.203 0.298 | | 0.176 0.276 | | 0.166 0.264 | | 0.193 0.295 | | 0.214 0.327 | |
| Traffic | **0.381** **0.283** | | 0.406 0.287 | | 0.408 0.291 | | 0.408 0.291 | | 0.414 0.295 | | 0.414 0.289 | | 0.412 0.289 | | 0.391 0.289 | | 0.384 0.295 | | 0.434 0.295 | | 0.620 0.336 | | 0.610 0.376 | |
| ETTh1 | **0.402** **0.420** | | 0.405 0.426 | | 0.414 0.435 | | 0.405 0.437 | | 0.418 0.431 | | 0.412 0.428 | | 0.416 0.428 | | 0.451 0.462 | | 0.429 0.437 | | 0.419 0.439 | | 0.520 0.503 | | 0.440 0.460 | |
| ETTh2 | **0.342** **0.383** | | 0.348 0.392 | | 0.355 0.398 | | 0.358 0.387 | | 0.367 0.402 | | 0.366 0.407 | | 0.351 0.392 | | 0.382 0.414 | | 0.367 0.393 | | 0.502 0.481 | | 0.425 0.451 | | 0.437 0.449 | |
| ETTm1 | **0.340** **0.371** | | 0.343 0.380 | | 0.350 0.383 | | 0.355 0.380 | | 0.355 0.386 | | 0.352 0.378 | | 0.355 0.381 | | 0.370 0.399 | | 0.388 0.385 | | 0.357 0.380 | | 0.400 0.418 | | 0.448 0.452 | |
| ETTm2 | **0.252** **0.311** | | 0.257 0.319 | | 0.272 0.332 | | 0.258 0.347 | | 0.264 0.328 | | 0.254 0.315 | | 0.263 0.322 | | 0.272 0.331 | | 0.277 0.326 | | 0.276 0.341 | | 0.305 0.355 | | 0.305 0.349 | |

**Results.** Table 1 summarizes the results of long-term time series forecasting. We can observe that: (1) MSH-LLM achieves the SOTA results in all datasets. Specifically, MSH-LLM achieves an average error reduction of 4.10% and 3.72% compared to LLM4TS methods (i.e., S$^2$IP-LLM, AutoTimes, Time-LLM, and FPT), 8.54% and 6.45% compared to the latest Transformer-based methods (i.e., ASHyper, iTransformer, and MSHyper), and 7.48% and 5.58% compared to the Linear-based methods (i.e., AMD and DLinear) in MSE and MAE, respectively. (2) Competitive performance is achieved by Ada-MSHyper, MSHyper, and PatchTST through the consideration of group-wise interactions. (3) LLM4TS methods (e.g., S$^2$IP-LLM and Time-LLM) further improve upon these by embedding group-wise interactions into LLMs. However, they overlook the multi-scale structures of natural language and time series. (4) By considering the multi-scale structures of natural language and time series, MSH-LLM outperforms other LLM4TS methods in almost all cases.

### 5.2 SHORT-TERM FORECASTING

**Setups.** To fully evaluate the performance of MSH-LLM, we also conduct short-term forecasting experiments on M4 datasets, which contain marketing data with different sampling frequencies. More details about M4 datasets are given in Appendix C. The forecasting lengths are set between 6 and 48, which are significantly shorter than those in long-term time series forecasting. Following existing works (Zhou et al., 2023a; Jin et al., 2024; Pan et al., 2024), we set the input length to be twice the

forecasting length. The symmetric mean absolute percentage error (SMAPE), mean absolute scaled error (MASE), and overall weighted average (OWA) are used as the evaluation metrics.

Table 2: The mean performance results for short-term time series forecasting on the M4 datasets. Smaller values correspond to better performance. **Bolded**: Best, Underlined: Second best. Full results are listed in Appendix F.2, Table 9.

| Methods | | MSH-LLM (Ours) | AutoTimes (NeurIPS 2024) | S²IP-LLM (ICML 2024) | Time-LLM (ICLR 2024) | FPT (NeurIPS 2023) | iTransformer (ICLR 2024) | DLinear (AAAI 2023) | PatchTST (ICLR 2023) | N-HiTS (AAAI 2023) | N-BEATS (ICLR 2020) | TimesNet (ICLR 2023) |
|---|---|---|---|---|---|---|---|---|---|---|---|---|
| | SMAPE | **11.659** | 11.831 | 12.021 | 12.494 | 12.690 | 12.142 | 13.639 | 12.059 | 12.035 | 12.25 | 12.88 |
| Avg. | MASE | **1.557** | 1.585 | 1.612 | 1.731 | 1.808 | 1.631 | 2.095 | 1.623 | 1.625 | 1.698 | 1.836 |
| | OWA | **0.837** | 0.850 | 0.857 | 0.913 | 0.940 | 0.874 | 1.051 | 0.869 | 0.869 | 0.896 | 0.955 |

**Results.** Table 2 gives the short-term time series forecasting results. We can see that: (1) MSH-LLM performs slightly better than AutoTimes and substantially exceeds other baseline methods. (2) By leveraging LLMs and Patch mechanisms, AutoTimes and PatchTST achieve competitive results than other baseline methods. (3) Compared to AutoTimes and PatchTST, MSH-LLM achieves superior performance, the reason may be that the hyperedging mechanism can enhance the multi-scale semantic information of time series semantic space while reducing irrelevant information interference.

## 5.3 TIME SERIES CLASSIFICATION

**Setups.** We also perform the time series classification task to verify the generalization ability of the model. Following existing works (Zhou et al., 2023a; Wu et al., 2022), we use 10 multivariate UEA time series classification datasets for evaluation, which cover different domains (e.g., gesture, medical diagnosis, and audio recognition). More details about the datasets are given in Appendix F.3. Accuracy is used as the evaluation metric.

**Results.** Figure 3 shows time series classification results. MSH-LLM achieves an average accuracy of 75.38%, surpassing all baselines including advanced LLM4TS methods FPT (74%). It is also notable that other methods considering multi-scale structures (e.g., TimesNet and Flowformer) can also achieve better performance.

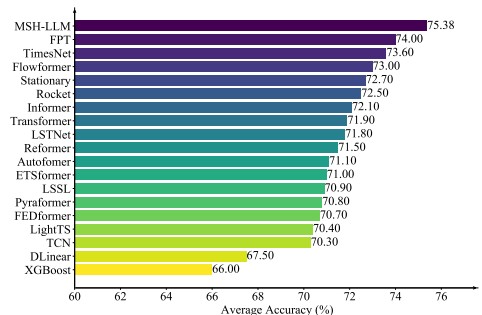

Figure 3: Time series classification results. The results are averaged from 10 subsets of UEA. Higher values mean better performance. Full results are given in Appendix F.3.

The reason is that the time series classification is a sequence-level task, and multi-scale structures help models learn hierarchical representations. However, MSH-LLM still performs better than those methods, the reason may be that MSH-LLM leverages MoP mechanism to enhance the reasoning capabilities of LLMs, thereby promoting LLMs to learn more comprehensive representations of multi-scale temporal patterns.

## 5.4 FEW-SHOT LEARNING

**Setups.** LLMs have shown impressive capabilities for few-shot learning (Liu et al., 2023a). Following existing works (Jin et al., 2024; Zhou et al., 2023a), we use limited training data (i.e., 5% and 10% of the training data) on 7 commonly used datasets to evaluate the few-shot learning performance.

Table 3: Few-shot learning results with 5% of the training data, averaged across all forecasting horizons. **Bolded**: Best, Underlined: Second best. Full results are given in Appendix F.4, Table 13.

| Methods | MSH-LLM (Ours) | S²IP-LLM (ICML 2024) | Time-LLM (ICLR 2024) | FPT (NeurIPS 2023) | iTransformer (ICLR 2024) | PatchTST (ICLR 2023) | TimesNet (ICLR 2023) | FEDformer (ICML 2022) | NSFormer (NeurIPS 2022) | ETSformer (arXiv 2022) | Autoformer (NeurIPS 2021) |
|---|---|---|---|---|---|---|---|---|---|---|---|
| Metric | MSE MAE | MSE MAE | MSE MAE | MSE MAE | MSE MAE | MSE MAE | MSE MAE | MSE MAE | MSE MAE | MSE MAE | MSE MAE |
| Weather | **0.247 0.281** | 0.260 0.297 | 0.264 0.301 | 0.263 0.301 | 0.309 0.339 | 0.269 0.303 | 0.298 0.318 | 0.309 0.353 | 0.310 0.353 | 0.327 0.328 | 0.333 0.371 |
| Electricity | **0.174 0.269** | 0.179 0.275 | 0.181 0.279 | 0.178 0.273 | 0.201 0.296 | 0.181 0.277 | 0.402 0.453 | 0.266 0.353 | 0.346 0.404 | 0.627 0.603 | 0.800 0.685 |
| Traffic | **0.413 0.292** | 0.420 0.299 | 0.423 0.302 | 0.434 0.305 | 0.450 0.324 | 0.418 0.296 | 0.867 0.493 | 0.676 0.423 | 0.833 0.502 | 1.526 0.839 | 1.859 0.927 |
| ETT(Avg) | **0.421 0.423** | 0.445 0.438 | 0.580 0.497 | 0.465 0.447 | 0.675 0.542 | 0.590 0.503 | 0.606 0.507 | 0.558 0.503 | 0.587 0.527 | 0.676 0.526 | 0.914 0.712 |

**Results.** Table 3 summarizes the few-shot learning results under 5% training data. We can see that MSH-LLM achieves SOTA results in almost all cases, reducing the average prediction error by 10.47% and 6.74% over other LLM4TS methods (i.e., $S^2$IP-LLM and Time-LLM) in terms of MSE and MAE, respectively. This may be attributed to the fact that MSH-LLM can consider the multi-scale structures of natural language and time series, while using the MoP mechanism to unlock the knowledge within LLMs to understand multi-scale patterns. The few-shot learning results under 10% training data are given in Appendix 5.4.

## 5.5 ZERO-SHOT LEARNING

**Setups.** In this section, we evaluate the performance of MSH-LLM for few-shot learning, where no training sample of the target domain is available. Specifically, we adhere to the benchmark established by (Zhou et al., 2023a; Liu et al., 2024) and evaluate the cross-dataset adaptation performance (i.e., the model's performs on dataset $A$ when trained on dataset $B$). M3 and M4 datasets are used to evaluate the zero-shot learning performance.

Table 4: Zero-shot learning results in terms of averaged SMAPE. **Bolded**: Best, Underlined: Second best. Full results are listed in Appendix F.5, Table 14.

| Methods | MSH-LLM (Ours) | AutoTimes (NeurIPS 2024) | FPT (NeurIPS 2023) | DLinear (AAAI 2023) | PatchTST (ICLR 2023) | TimesNet (ICLR 2023) | NSformer (NeurIPS 2022) | FEDformer (ICML 2022) | Informer (AAAI 2021) | Reformer (ICLR 2019) |
|---|---|---|---|---|---|---|---|---|---|---|
| M4→M3 | **12.469** | 12.750 | 13.060 | 14.030 | 13.390 | 14.170 | 15.290 | 13.530 | 15.820 | 13.370 |
| M3→M4 | **12.968** | 13.036 | 13.125 | 15.337 | 13.228 | 14.553 | 14.327 | 15.047 | 19.047 | 14.092 |

**Results.** Table 4 provides the zero-shot learning results. MSH-LLM consistently outperforms other methods in zero-shot scenarios. Both M3 and M4 pose significant challenges due to their sophisticated multi-scale characteristics and diverse domain distributions. MSH-LLM still achieves the best performance, giving an average of 10.23% SMAPE error reductions across all baselines on average. This improvement may stem from its ability to leverage the reasoning capabilities of LLMs for interpreting multi-scale temporal patterns.

## 5.6 ABLATION STUDIES

**LLMs Selection.** Scaling law is an essential characteristic that extends from small models to large foundation models. To investigate the impact of backbone model size, we design the following three variants: (1) Using the first 12 Transformer layers of LLaMA-7B (**L.12**). (2) Replacing LLaMA-7B with GPT-2 Small (**G.12**). (3) Replacing LLaMA-7B with the first 6 Transformer layers of GPT-2 Small (**G.6**). The experimental results on the Traffic dataset are shown in Table 5. We can observe that MSH-LLM (Default 32) performs better than **L.12**, **G.12**, and **G.6**, which indicate that the scaling law also applies to cross-modalities alignment with frozen LLMs.

Table 5: Results of different LLMs selection and MoP mechanism. The best results are **bolded**.

| Methods | L.12 | G.12 | G.6 | -w/o $\mathcal{C}_l$ | -w/o $\mathcal{C}_d$ | -w/o $\mathcal{C}_c$ | -w/o MoP | MSH-LLM (**Default**:32) |
|---|---|---|---|---|---|---|---|---|
| Metric | MSE MAE | MSE MAE | MSE MAE | MSE MAE | MSE MAE | MSE MAE | MSE MAE | MSE MAE |
| 96 | 0.370 0.274 | 0.377 0.281 | 0.393 0.295 | 0.373 0.272 | 0.368 0.273 | 0.375 0.272 | 0.399 0.283 | **0.365 0.270** |
| 192 | 0.375 0.283 | 0.385 0.290 | 0.404 0.297 | 0.379 0.289 | 0.383 0.286 | 0.392 0.282 | 0.403 0.290 | **0.372 0.281** |
| 336 | 0.393 0.286 | 0.397 0.289 | 0.411 0.316 | 0.400 0.293 | 0.405 0.292 | 0.391 0.284 | 0.409 0.295 | **0.385 0.279** |

**MoP Mechanism.** To investigate the impact of MoP mechanism, we design three variants: (1) Removing the learnable prompts (-w/o $\mathcal{C}_l$). (2) Removing the data-correlated prompts (-w/o $\mathcal{C}_d$). (3) Removing the capability-enhancing prompts (-w/o $\mathcal{C}_c$). (3) Removing the MoP mechanism (-w/o **MoP**). The experimental results on the Traffic dataset are shown in Table 5, from which we can observe that MSH-LLM performs better than -w/o $\mathcal{C}_l$, -w/o $\mathcal{C}_d$, and -w/o $\mathcal{C}_c$, showing the effectiveness of learnable prompts, data-correlated prompts, and capability-enhancing prompts, respectively. In addition, -w/o **MoP** achieves the worst performance, demonstrating the effectiveness of the MoP mechanism. More ablation experiments on the MoP mechanism, hyperedging mechanism, ME module, and CMA module are shown in Appendix G and H.

## 5.7 VISUALIZATION

**Visualization of the MoP Mechanism.** To investigate how prompts can guide LLMs in time series analysis. The t-SNE results on the Traffic dataset are provided in Figure 4. We can observe that the output of LLM with the MoP mechanism (Figure 4(a)) shows distinct clusters, while the output of LLM without the MoP mechanism (Figure 4(e)) reveals a more spread-out and lacks clear clustering. The experimental results show the effectiveness of the MoP mechanism in activating the abilities of LLMs to capture multi-scale temporal patterns. In addition, we observe that Figures 4(a) and Figure 4(b) (-w/o $\mathcal{C}_d$) share similar clusters, while Figures 4(c) (-w/o $\mathcal{C}_l$) and Figure 4(d) (-w/o $\mathcal{C}_c$) show less distinct clusters compared to Figure 4(a), suggesting that $\mathcal{C}_d$ has a relatively minor influence compared $\mathcal{C}_l$ and $\mathcal{C}_c$ on the Traffic dataset for long-term forecasting. However, this does not imply that $\mathcal{C}_d$ is unimportant, as removing $\mathcal{C}_d$ leads to a performance degradation. The qualitative analysis also aligns with the experimental results in Table 5.

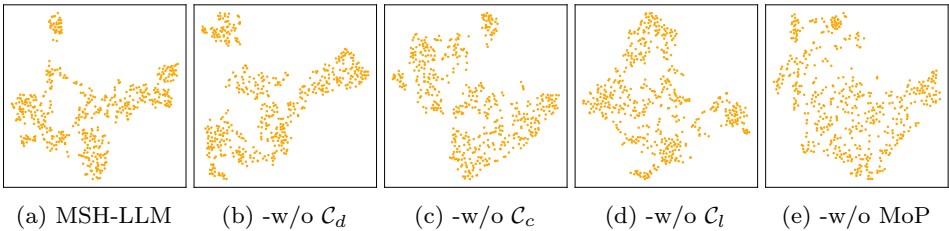

(a) MSH-LLM      (b) -w/o $\mathcal{C}_d$      (c) -w/o $\mathcal{C}_c$      (d) -w/o $\mathcal{C}_l$      (e) -w/o MoP

Figure 4: t-SNE visualization of pre-trained LLM outputs under different prompts.

**Visualization of the Weight Between Text Prototypes and Word Embeddings.** To investigate whether different text prototypes possess explicit semantic meanings, we conduct qualitative analysis by visualizing the similarity scores between 10 randomly selected text prototypes and word embeddings derived from 3 different word sets. The visualization results on the ETTh1 dataset are given in Figure 5. We can discern the following tendencies: 1) Prototypes 2, 3, 7, and 8 exhibit strong associations with word set 1 (noun-like time series descriptions), while prototypes 0, 1, and 4 show strong correlations with word set 2 (adjective-like time series descriptions). This suggests that the prototypes capture different semantic roles, indicating explicit semantic differentiation. 2) Although both word set 1 and word set 3 consist of noun-like descriptions, almost all prototypes show weak correlations with word set 3 (name-related words). The reason may be that the text prototypes encode time-series-specific, context-specific semantic information. The experimental results show that the text prototypes possess explicit meaning.

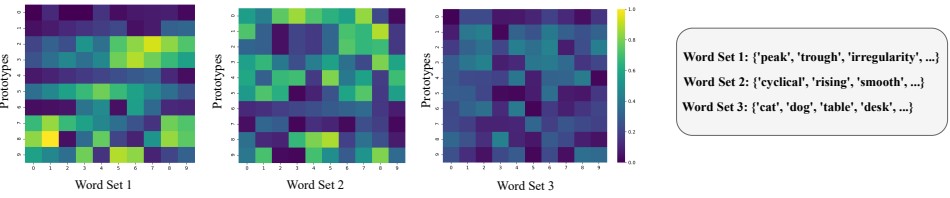

Figure 5: The visualization of the weight between text prototypes and word embeddings.

## 6 CONCLUSIONS

In this paper, we present MSH-LLM, a multi-scale hypergraph framework that aligns pre-trained large language models for time series analysis. Empowered by the hyperedging mechanism and cross-modality alignment (CMA) module, MSH-LLM can perform alignment at different scales, addressing the problem of multi-scale semantic space disparity between natural language and time series. Then, a mixture of prompts (MoP) mechanism is introduced to enhance the reasoning capabilities of LLMs towards multi-scale temporal patterns. Experimental results on 27 real-world datasets across 5 different applications demonstrate that MSH-LLM consistently outperforms existing methods.

## 7 ACKNOWLEDGEMENT

This work was funded by the National Key Research and Development Program of China (No.2024YFB3312900).

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

# A  DESCRIPTION OF BASELINES

To evaluate the effectiveness of MSH-LLM, we select 17 advanced baselines for comparison. The detailed descriptions of these baselines are given as follows:

**AutoTimes** (Liu et al., 2024): AutoTimes repurposes LLMs as autoregressive time series forecasters by projecting time series into language token embeddings and utilizing in-context forecasting.

**Time-LLM** (Jin et al., 2024): Time-LLM is a reprogramming framework that repurposes large language models (LLMs) for time series forecasting by aligning time series data with text prototypes and enhancing the model's reasoning ability through a Prompt-as-Prefix (PaP) mechanism.

**FPT** (Zhou et al., 2023a): FPT leverages pre-trained language and CV models for time series analysis by fine-tuning them without altering their self-attention and feedforward layers.

**$S^2$IP-LLM** (Pan et al., 2024): $S^2$IP-LLM aligns the pre-trained semantic space of LLMs with time series embeddings by designing a cross-modality tokenization module and maximizing cosine similarity between semantic anchors and time series components, enabling improved time series forecasting through learned prompts.

**DLinear** (Zeng et al., 2022): DLinear uses the simple one-layer model to capture temporal dependencies of different decomposed components.

**N-HiTS** (Challu et al., 2023): N-HiTS proposes a novel hierarchical interpolation and multi-rate data sampling techniques to model multi-scale temporal patterns.

**N-BEATS** (Oreshkin et al., 2020): N-BEATS employs a deep stack of fully-connected layers to capture temporal dependencies.

**AMD** (Hu et al., 2025): AMD decomposes time series into distinct temporal patterns at different scales and leverages the multi-scale decomposable mixing block to dissect and aggregate these patterns in a residual manner.

**Ada-MSHyper** (Shang et al., 2024a): Ada-MSHyper utilizes an adaptive hypergraph to capture multi-scale group-wise interactions, while introducing the node and hyperedge constraint mechanism to address the issue of entangled temporal variations.

**iTransformer** (Liu et al., 2023b): iTransformer inverts the Transformer's input dimensions, applying attention and feed-forward networks on variate tokens to capture cross-variate correlations and improve long-sequence forecasting.

**PatchTST** (Nie et al., 2022): PatchTST proposes a Transformer-based model for multivariate time series forecasting by segmenting time series into subseries-level patches and applying channel-independence strategies.

**TimesNet** (Wu et al., 2022): TimesNet models complex temporal variations in time series by transforming 1D series into 2D tensors.

**MSHyper** (Shang et al., 2024b): MSHyper combines the multi-scale feature extraction module with the pre-defined multi-scale hypergraph for group-wise pattern interactions.

**Autoformer** (Wu et al., 2021): Autoformer introduces a decomposition architecture with an Auto-Correlation mechanism to capture the long-range dependencies.

**NSFormer** (Liu et al., 2022): NSFormer employs a set of stationarization modules and the de-stationary attention to tackle the problem of over-stationarization.

**FEDformer** (Zhou et al., 2022): FEDformer improves long-term time series forecasting by combining seasonal-trend decomposition with a frequency-enhanced Transformer, capturing both global patterns and detailed structures efficiently with linear complexity.

**ETSformer** (Woo et al., 2022b): ETSformer incorporates the principles of exponential smoothing by replacing self-attention with exponential smoothing and frequency attention for time series forecasting.

**Reformer** (Kitaev et al., 2019): Reformer improves Transformer efficiency by replacing dot-product attention with locality-sensitive hashing and using reversible residual layers.

**Informer** (Zhou et al., 2021): Informer introduces a ProbSparse self-attention mechanism, self-attention distilling, and a generative-style decoder for long-sequence time series forecasting, addressing time complexity, memory issues, and improving prediction efficiency and accuracy.

## B  DESCRIPTION OF HYPERGRAPH LEARNING

Compared to Graph Neural Networks (GNNs) (Chen et al., 2023a; Chen & Zhao, 2025; Chen & Chen, 2024), which model pairwise interactions by operating on graphs where each edge connects exactly two nodes, Hypergraph Neural Networks (HGNNs) (Shang et al., 2024a;b; Wu et al., 2025a) generalize this paradigm to capture group-wise interactions through hyperedges that can connect an arbitrary number of nodes. As shown in Figure 6, the graph is represented using the adjacency matrix, in which each edge connects two nodes. In contrast, the hypergraph is represented by the incidence matrix, which can capture group-wise interaction using its degree-free hyperedges.

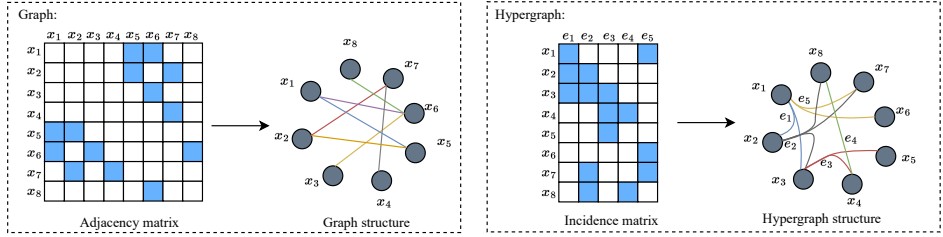

Figure 6: The comparison between graph and hypergraph.

Recently, HGNNs have been applied in different fields, e.g., video object segmentation (Huang et al., 2009), stock selection (Sawhney et al., 2021), temporal knowledge graphs (Tang et al., 2024), and time series forecasting (Shang et al., 2024a;b; Zhao et al., 2023; Tian et al., 2025). HyperGCN (Yadati et al., 2019) is the first work to incorporate convolutional operations into hypergraphs, demonstrating the superiority of HGNNs over ordinary GNNs in capturing group-wise interactions. STHAN-SR (Sawhney et al., 2021) reformulates stock prediction as a learning-to-rank task and utilizes hypergraphs to capture group-wise interactions between stocks. GroupNet (Xu et al., 2022) employs multi-scale hypergraphs for trajectory prediction, which combines relational reasoning with hypergraph structures to capture group-wise pattern interactions among multiple agents. In the context of time series forecasting, MSHyper (Shang et al., 2024b) is the first work to incorporate hypergraphs into long-term time series forecasting, which leverages predefined hypergraphs and the tri-stage message passing mechanism to capture multi-scale pattern interactions. Building on this, Ada-MSHyper (Shang et al., 2024a) introduces adaptive hypergraph modeling, which combines adaptive hypergraphs with the node and hyperedge constraint mechanism to capture abundant and implicit group-wise temporal pattern interactions.

In this paper, we represent temporal features of different scales as nodes and use learnable hyperedges in the hypergraph to capture group-wise interactions, thereby enhancing the semantic information of time series semantic space. We formulate this process as the hyperedging mechanism. As mentioned above, our hyperedging mechanism differs from previous methods in two aspects. Firstly, our methods can capture implicit group-wise interactions at different scales in a learnable manner, while most existing methods (Nie et al., 2022; Zhou et al., 2023a; Shang et al., 2024b) rely on pre-defined rules to model group-wise interactions at a single scale. Secondly, although some methods (Shang et al., 2024a; Jiang et al., 2019) learn from hypergraphs, they focus on constraints or clustering-based approaches to learn the hypergraph structures. In contrast, our method learns the hypergraph structures in a pure data-driven manner by incorporating learnable parameters and nonlinear transformations, which is more flexible and can learn more complex hypergraph structures.

## C  DESCRIPTION OF DATASETS

To evaluate the performance of MSH-LLM, we use 7 real-world datasets for long-term time series forecasting and few-shot learning. Detailed description of these detasets are given as follows:

- **Electricity Transformer Temperature (ETT) datasets**: These datasets are collected from power grid snapshots in two distinct regions of China. They encompass four subsets: ETTh1 and ETTh2 with an hourly sampling rate, and ETTm1 and ETTm2 with a 15-minute sampling rate.

- **Traffic dataset**[1]: The Traffic dataset records the highway occupancy rates via 862 individual sensors. It is collected at a 1-hour granularity, forming a complex multivariate time series for evaluating long-term dependencies.

- **Electricity dataset**[2]: The Electricity dataset contains the hourly electricity consumption of 321 individual clients. It is recorded at a 1-hour granularity.

- **Weather datasets**[3]: The Weather dataset records 21 meteorological indicators collected from weather stations in Germany, with observations recorded at a 10-minute granularity throughout the year 2020.

In addition, we employ M4 datasets (including M4 Yearly, M4 Quarterly, M4 Monthly, M4 Weekly, M4 Daily, and M4 Hourly) for short-term forecasting and both M3 (including M3 Yearly, M3 Quarterly, M3 Monthly, and M3 Others) and M4 datasets for zero-shot learning. Detailed descriptions of M3 and M4 datasets are given as follows:

- **M3 datasets**: The M3 datasets comprise 3,003 time series with varying granularities, including micro, industry, macro, and finance. It presents a diverse collection of seasonal and non-seasonal sequences, with frequencies spanning from yearly to monthly.

- **M4 datasets**: The M4 datasets are larger than M3 datasets and expands the collection to 100,000 individual time series. The M4 datasets encompass a vast array of temporal patterns across six distinct frequencies (ranging from hourly to yearly).

Finally, we use 10 multivariate datasets selected from the UEA time series classification Archive (Bagnall et al., 2018; Zerveas et al., 2021) for time series classification. These datasets are complex, which cover different domains (e.g., gesture, medical diagnosis, and audio recognition) and exhibiting diverse characteristics in terms of sample size, dimensionality, and number of classes. The detailed descriptions of the datasets are provided in Table 6.

Table 6: Dataset descriptions for time series classification. The dataset size is organized in (training, validation, and testing).

| Dataset | Dataset size | Variates | Classes | Information |
|---|---|---|---|---|
| EthanolConcentration | (261, 0, 263) | 3 | 4 | Biomedical |
| FaceDetection | (5890, 0, 3524) | 144 | 2 | Computer Vision |
| Handwriting | (150, 0, 850) | 3 | 26 | Pattern Recognition |
| Heartbeat | (204, 0, 205) | 61 | 2 | Medical Recognition |
| JapaneseVowels | (270, 0, 370) | 12 | 9 | Audio Recognition |
| PEMS-SF | (267, 0, 173) | 963 | 7 | Transportation |
| SelfRegulationSCP1 | (268, 0, 293) | 6 | 2 | Psychology |
| SelfRegulationSCP2 | (200, 0, 180) | 7 | 2 | Psychology |
| SpokenArabicDigits | (6599, 0, 2199) | 13 | 10 | Speech Recognition |
| UWaveGestureLibrary | (120, 0, 320) | 3 | 8 | Gesture |

We follow the same data processing and training-validation-testing split protocol as in existing works (Zhou et al., 2023a; Jin et al., 2024; Pan et al., 2024; Wu et al., 2025b; 2024; Liu et al., 2024). For data partitioning, we adopt split ratios of 6:2:2 for the ETT datasets and 7:2:1 for the Electricity, Traffic, and Weather datasets. For the few-shot learning task, only 5% or 10% of the original training data is retained, while the validation and testing sets stay the same.

---

[1] http://pems.dot.ca.gov
[2] https://archive.ics.uci.edu/ml/datasets/ElectricityLoadDiagrams20112014
[3] https://www.bgc-jena.mpg.de/wetter/

## D    EXPERIMENTAL SETTINGS

MSH-LLM is implemented using the PyTorch library(Paszke et al., 2019). The number of scales $S$ is set to 3, and we employ 1D convolution as the aggregation function. All experimental evaluations are carried out on NVIDIA A100 (80GB) and NVIDIA GeForce RTX 3090 GPUs. LLaMA-7B (Touvron et al., 2023) is used as the default base LLM. We use the Adam optimizer (Kingma, 2014) with an initial learning rate selected from $\{10^{-3}, 5 \times 10^{-3}, 10^{-4}\}$. Unlike prior works that rely on grid search for hyperparameter tuning, we leverage the Neural Network Intelligence (NNI) toolkit [4] to automatically optimize the hyperparameters. The specific search space for the hyperparameters is detailed in Table 7. The source code for MSH-LLM is publicly available on GitHub [5].

Table 7: The search space of hyperparameters.

| Parameters | Choise |
|---|---|
| Batch size | {8, 16, 32, 64, 128, 256} |
| Number of hyperedges at scale 1 | {5, 10, 20, 30, 50} |
| Number of hyperedges at scale 2 | {2, 5, 10, 15, 20} |
| Number of hyperedges at scale 3 | {1, 2, 4, 5, 8, 12} |
| Number of text prototypes at scale 1 | {20, 50, 100, 200, 500, 1000} |
| Number of text prototypes at scale 1 | {10, 25, 50, 100, 200, 500} |
| Number of text prototypes at scale 1 | {4, 5, 10, 25, 50, 100} |
| Aggregation window size at scale 1 | {2, 4, 8} |
| Aggregation window size at scale 2 | {2, 4} |
| $\eta$ | {2, 3, 4, 5, 10, 15, 20} |

## E    EVALUATION METRICS

For long-term time series forecasting and few-shot learning, we employ the Mean Squared Error (MSE) and Mean Absolute Error (MAE) as our evaluation metrics, which can be formulated as follows:

$$\text{MSE} = \frac{1}{H} \left\| \widehat{\mathbf{X}}^{\text{O}}_{T+1:T+H} - \mathbf{X}^{\text{O}}_{T+1:T+H} \right\|_2^2, \quad \text{MAE} = \frac{1}{H} |\widehat{\mathbf{X}}^{\text{O}}_{T+1:T+H} - \mathbf{X}^{\text{O}}_{T+1:T+H}|, \quad (10)$$

where $T$ and $H$ are the input and output lengths, $\widehat{\mathbf{X}}^{\text{O}}_{T+1:T+H}$ and $\mathbf{X}^{\text{O}}_{T+1:T+H}$ are the forecasting results and ground truth, respectively.

For short-term time series forecasting and zero-shot learning tasks, the Symmetric Mean Absolute Percentage Error (SMAPE), Overall Weighted Average (OWA), and Mean Absolute Scaled Error (MASE) are employed as evaluation metrics, which are defined as follows:

$$\text{SMAPE} = \frac{200}{H} \sum_{h=1}^{H} \frac{|\widehat{\mathbf{X}}^{\text{O}}_{T+1:T+H} - \mathbf{X}^{\text{O}}_{T+1:T+H}|}{|\mathbf{X}^{\text{O}}_{T+1:T+H}|}, \quad (11)$$

$$\text{OWA} = \frac{1}{2} \left[ \frac{\text{SMAPE}}{\text{SMAPE}_{\text{Naive2}}} + \frac{\text{MASE}}{\text{MASE}_{\text{Naive2}}} \right], \quad (12)$$

$$\text{MASE} = \frac{1}{H} \sum_{h=1}^{H} \frac{|\widehat{\mathbf{X}}^{\text{O}}_{T+1:T+H} - \mathbf{X}^{\text{O}}_{T+1:T+H}|}{\frac{1}{H-s} \sum_{j=s+1}^{H} |\mathbf{X}^{\text{O}}_{T+1:T+H} - \mathbf{X}^{\text{O}}_{T+1:T+H-1}|}, \quad (13)$$

## F    FULL RESULTS

To ensure a fair comparison, we adhere to the unified experimental settings used in previous works (Zhou et al., 2023a; Pan et al., 2024; Jin et al., 2024). The average results represent the mean of

---

[4] https://nni.readthedocs.io/en/latest/
[5] https://github.com/shangzongjiang/MSH-LLM/

outcomes across different forecasting scenarios, with the best results **bolded** and the second-best results underlined. An asterisk (*) indicates that some results do not align with the unified settings, so we reran their official code under the unified conditions and fine-tuned their key hyperparameters.

## F.1 LONG-TERM TIME SERIES FORECASTING

Table 8 summarizes the full results of long-term time series forecasting. We can observe that MSH-LLM achieves the SOTA results in 54 out of 70 cases across 7 time series datasets. Specifically, on the well-studied Traffic dataset, MSH-LLM achieves an average error reduction of 11.54% and 6.71% across all baselines. On the challenging Weather dataset, MSH-LLM achieves an average error reduction of 12.78% and 11.26% across all baselines.

Table 8: Full results of long-term time series forecasting. The input length is set to 512, and the forecasting lengths are set to 96, 192, 336, and 720. Smaller values correspond to better performance. **Bolded**: Best, Underlined: Second best.

| Methods | | MSH-LLM (Ours) | S²IP-LLM (ICLR 2024) | Time-LLM (ICLR 2024) | AutoTimes* (NeurIPS 2024) | FPT (NeurIPS 2023) | AMD* (AAAI 2025) | ASHyper* (NeurIPS 2024) | iTransformer (ICLR 2024) | MSHyper* (arXiv 2024) | DLinear (AAAI 2023) | TimesNet (ICLR 2023) | FEDformer (ICML 2022) |
|---|---|---|---|---|---|---|---|---|---|---|---|---|---|
| Metric | | MSE MAE | MSE MAE | MSE MAE | MSE MAE | MSE MAE | MSE MAE | MSE MAE | MSE MAE | MSE MAE | MSE MAE | MSE MAE | MSE MAE |
| Weather | 96 | **0.138 0.187** | 0.145 0.195 | 0.158 0.210 | 0.161 0.216 | 0.162 0.212 | 0.148 0.203 | 0.169 0.228 | 0.253 0.304 | 0.171 0.212 | 0.176 0.237 | 0.172 0.220 | 0.217 0.296 |
| | 192 | **0.187 0.230** | 0.190 0.235 | 0.197 0.245 | 0.205 0.253 | 0.204 0.248 | 0.193 0.243 | 0.235 0.288 | 0.280 0.319 | 0.214 0.250 | 0.220 0.282 | 0.219 0.261 | 0.276 0.336 |
| | 336 | **0.237** 0.282 | 0.243 **0.280** | 0.248 0.285 | 0.251 0.289 | 0.254 0.286 | 0.242 0.281 | 0.275 0.287 | 0.321 0.344 | 0.260 0.287 | 0.265 0.319 | 0.280 0.306 | 0.339 0.380 |
| | 720 | **0.305 0.315** | 0.312 0.326 | 0.319 0.334 | 0.314 0.356 | 0.326 0.337 | 0.315 0.332 | 0.335 0.327 | 0.364 0.374 | 0.327 0.336 | 0.333 0.362 | 0.365 0.359 | 0.403 0.428 |
| | Avg. | **0.217 0.254** | 0.223 0.259 | 0.231 0.269 | 0.233 0.279 | 0.237 0.271 | 0.225 0.265 | 0.254 0.283 | 0.305 0.335 | 0.243 0.271 | 0.249 0.300 | 0.259 0.287 | 0.309 0.360 |
| Electricity | 96 | **0.127** 0.231 | 0.135 0.230 | 0.137 0.237 | 0.134 0.233 | 0.139 0.238 | 0.131 **0.228** | 0.129 0.234 | 0.147 0.248 | 0.147 0.251 | 0.140 0.237 | 0.168 0.272 | 0.193 0.308 |
| | 192 | 0.150 0.242 | 0.149 0.247 | 0.150 0.249 | 0.150 0.247 | 0.153 0.251 | 0.151 0.244 | 0.154 0.227 | 0.165 0.267 | 0.167 0.269 | 0.153 0.249 | 0.184 0.289 | 0.201 0.315 |
| | 336 | **0.162 0.258** | 0.167 0.266 | 0.168 0.266 | 0.165 0.264 | 0.169 0.266 | 0.167 0.262 | 0.165 0.262 | 0.178 0.279 | 0.174 0.275 | 0.169 0.267 | 0.198 0.300 | 0.214 0.329 |
| | 720 | **0.198 0.279** | 0.200 0.287 | 0.203 0.293 | 0.203 0.293 | 0.199 0.298 | 0.206 0.297 | 0.200 0.292 | 0.201 0.290 | 0.322 0.398 | 0.216 0.308 | 0.203 0.301 | 0.220 0.320 | 0.246 0.355 |
| | Avg. | **0.159 0.253** | 0.163 0.258 | 0.165 0.261 | 0.165 0.261 | 0.167 0.263 | 0.162 0.257 | 0.162 0.253 | 0.203 0.298 | 0.176 0.276 | 0.166 0.264 | 0.193 0.295 | 0.214 0.327 |
| Traffic | 96 | **0.365 0.270** | 0.379 0.274 | 0.380 0.277 | 0.366 0.279 | 0.388 0.282 | 0.387 0.278 | 0.368 0.277 | 0.367 0.288 | 0.394 0.389 | 0.410 0.282 | 0.593 0.321 | 0.587 0.366 |
| | 192 | **0.372 0.281** | 0.397 0.282 | 0.399 0.288 | 0.395 0.287 | 0.407 0.290 | 0.402 0.282 | 0.379 0.288 | 0.378 0.293 | 0.375 0.289 | 0.395 0.287 | 0.617 0.336 | 0.604 0.373 |
| | 336 | **0.385 0.279** | 0.407 0.289 | 0.408 0.290 | 0.406 0.283 | 0.412 0.294 | 0.413 0.288 | 0.397 0.292 | 0.389 0.294 | 0.395 0.283 | 0.436 0.296 | 0.629 0.336 | 0.621 0.383 |
| | 720 | **0.402** 0.303 | 0.440 0.301 | 0.445 0.308 | 0.421 0.305 | 0.450 0.312 | 0.444 0.306 | 0.421 **0.298** | 0.401 0.304 | 0.407 0.308 | 0.466 0.315 | 0.640 0.350 | 0.626 0.382 |
| | Avg. | **0.381 0.283** | 0.406 0.287 | 0.408 0.291 | 0.397 0.289 | 0.414 0.295 | 0.412 0.289 | 0.391 0.289 | 0.384 0.295 | 0.393 0.317 | 0.434 0.295 | 0.620 0.336 | 0.610 0.376 |
| ETTh1 | 96 | **0.360 0.388** | 0.366 0.396 | 0.383 0.410 | 0.368 0.395 | 0.379 0.402 | 0.371 0.399 | 0.368 0.391 | 0.395 0.420 | 0.372 0.417 | 0.367 0.396 | 0.468 0.475 | 0.376 0.419 |
| | 192 | **0.398 0.411** | 0.401 0.420 | 0.419 0.435 | 0.404 0.415 | 0.415 0.424 | 0.403 0.420 | 0.429 0.417 | 0.427 0.441 | 0.418 0.432 | 0.401 0.419 | 0.484 0.485 | 0.420 0.448 |
| | 336 | 0.415 0.432 | 0.412 0.431 | 0.426 0.440 | 0.408 0.435 | 0.435 0.440 | 0.423 0.432 | 0.419 0.438 | 0.445 0.457 | 0.451 0.440 | 0.434 0.449 | 0.536 0.516 | 0.459 0.465 |
| | 720 | **0.436 0.447** | 0.440 0.458 | 0.440 0.458 | 0.428 0.456 | 0.439 0.503 | 0.441 0.459 | 0.452 0.461 | 0.446 0.465 | 0.537 0.530 | 0.476 0.458 | 0.472 0.493 | 0.593 0.537 | 0.506 0.507 |
| | Avg. | **0.402 0.420** | 0.405 0.426 | 0.414 0.435 | 0.405 0.426 | 0.418 0.431 | 0.412 0.428 | 0.416 0.428 | 0.451 0.462 | 0.429 0.437 | 0.419 0.439 | 0.502 0.503 | 0.440 0.460 |
| ETTh2 | 96 | **0.273** 0.331 | 0.278 0.340 | 0.297 0.357 | 0.282 0.329 | 0.289 0.347 | 0.279 0.343 | 0.274 0.335 | 0.304 0.360 | 0.287 0.331 | 0.301 0.367 | 0.376 0.415 | 0.358 0.397 |
| | 192 | **0.335 0.372** | 0.346 0.385 | 0.349 0.390 | 0.352 0.391 | 0.358 0.392 | 0.363 0.397 | 0.352 0.377 | 0.377 0.403 | 0.372 0.389 | 0.394 0.427 | 0.409 0.440 | 0.429 0.439 |
| | 336 | **0.363 0.400** | 0.367 0.406 | 0.373 0.408 | 0.382 0.403 | 0.383 0.414 | 0.381 0.419 | 0.369 0.427 | 0.405 0.429 | 0.407 0.423 | 0.506 0.495 | 0.425 0.455 | 0.496 0.487 |
| | 720 | **0.396 0.428** | 0.400 0.436 | 0.400 0.436 | 0.417 0.425 | 0.438 0.456 | 0.442 0.467 | 0.407 0.430 | 0.443 0.464 | 0.400 0.428 | 0.805 0.635 | 0.488 0.494 | 0.463 0.474 |
| | Avg. | **0.342 0.383** | 0.348 0.392 | 0.355 0.398 | 0.358 0.387 | 0.367 0.402 | 0.366 0.407 | 0.351 0.392 | 0.382 0.414 | 0.367 0.393 | 0.502 0.481 | 0.425 0.451 | 0.437 0.449 |
| ETTm1 | 96 | **0.285** 0.340 | 0.288 0.346 | 0.291 0.346 | 0.301 0.347 | 0.296 0.353 | 0.289 0.343 | 0.297 **0.338** | 0.312 0.366 | 0.323 0.348 | 0.323 0.348 | 0.329 0.377 | 0.379 0.419 |
| | 192 | **0.313 0.358** | 0.323 0.365 | 0.336 0.373 | 0.331 0.371 | 0.335 0.373 | 0.329 0.366 | 0.333 0.367 | 0.347 0.385 | 0.368 0.369 | 0.336 0.367 | 0.371 0.401 | 0.426 0.441 |
| | 336 | **0.355 0.377** | 0.359 0.390 | 0.362 0.390 | 0.365 0.380 | 0.369 0.394 | 0.365 0.386 | 0.365 0.388 | 0.379 0.404 | 0.392 0.390 | 0.368 0.387 | 0.417 0.428 | 0.445 0.459 |
| | 720 | **0.405 0.410** | 0.403 0.418 | 0.410 0.421 | 0.423 0.422 | 0.418 0.424 | 0.423 0.417 | 0.425 0.431 | 0.441 0.442 | 0.469 0.433 | 0.421 0.418 | 0.483 0.464 | 0.543 0.490 |
| | Avg. | **0.340 0.371** | 0.343 0.380 | 0.350 0.383 | 0.355 0.380 | 0.355 0.386 | 0.352 0.372 | 0.355 0.381 | 0.370 0.399 | 0.388 0.385 | 0.357 0.380 | 0.400 0.418 | 0.448 0.452 |
| ETTm2 | 96 | **0.161 0.246** | 0.165 0.257 | 0.184 0.275 | 0.167 0.261 | 0.170 0.264 | 0.168 0.258 | 0.168 0.256 | 0.179 0.271 | 0.242 0.313 | 0.243 0.311 | 0.168 0.263 | 0.201 0.286 | 0.203 0.287 |
| | 192 | **0.218 0.284** | 0.222 0.299 | 0.238 0.310 | 0.214 0.311 | 0.231 0.306 | 0.221 0.295 | 0.229 0.301 | 0.242 0.313 | 0.243 0.311 | 0.229 0.310 | 0.269 0.328 |
| | 336 | **0.271 0.320** | 0.277 0.330 | 0.286 0.340 | 0.284 0.325 | 0.280 0.339 | 0.271 0.327 | 0.281 0.334 | 0.288 0.344 | 0.299 0.338 | 0.289 0.352 | 0.331 0.376 | 0.325 0.366 |
| | 720 | **0.358** 0.392 | 0.363 0.390 | 0.379 0.403 | 0.367 0.492 | 0.373 0.402 | 0.355 0.381 | 0.372 0.397 | 0.378 0.397 | 0.397 0.399 | 0.416 0.437 | 0.428 0.430 | 0.421 0.415 |
| | Avg. | **0.252 0.311** | 0.257 0.319 | 0.272 0.332 | 0.258 0.347 | 0.264 0.328 | 0.254 0.315 | 0.263 0.322 | 0.272 0.331 | 0.277 0.326 | 0.276 0.341 | 0.305 0.355 | 0.305 0.349 |

## F.2 SHORT-TERM TIME SERIES FORECASTING

Table 9 summarizes the full results of short-term time series forecasting. The results on M4 Others are averaged from M4 Weekly, M4 Daily, and M4 Hourly. We can observe that MSH-LLM achieves the SOTA results on almost all datasets. Specifically, MSH-LLM performs slightly better than AutoTimes and S²IP-LLM (i.e., 1.45% and 3.01% average SMAPE improvement), outperforming other latest baselines by a large margin (e.g., 6.68% and 8.13% average SMAPE improvement over Time-LLM and FPT, respectively).

Table 9: Full results of short-term time series forecasting. We follow the protocol of existing work (Pan et al., 2024) and set the input length to twice the output length. Smaller values correspond to better performance. **Bolded**: Best, Underlined: Second best.

| Methods | | MSH-LLM (Ours) | AutoTimes* (NeurIPS 2024) | S²IP-LLM (ICML 2024) | Time-LLM (ICLR 2024) | FPT (NeurIPS 2023) | iTransformer (ICLR 2024) | DLinear (AAAI 2023) | PatchTST (ICLR 2023) | N-HiTS (AAAI 2023) | N-BEATS (ICLR 2020) | TimesNet (ICLR 2023) |
|---|---|---|---|---|---|---|---|---|---|---|---|---|
| Year. | SMAPE | **13.305** | 13.319 | 13.413 | 13.750 | 15.110 | 13.652 | 16.965 | 13.477 | 13.422 | 13.487 | 15.378 |
| | MASE | 2.995 | **2.993** | 3.024 | 3.055 | 3.565 | 3.095 | 4.283 | 3.019 | 3.056 | 3.036 | 3.554 |
| | OWA | **0.784** | **0.784** | 0.792 | 0.805 | 0.911 | 0.807 | 1.058 | 0.792 | 0.795 | 0.795 | 0.918 |
| Quart. | SMAPE | **10.024** | 10.101 | 10.352 | 10.671 | 10.597 | 10.353 | 12.145 | 10.380 | 10.185 | 10.564 | 10.465 |
| | MASE | **1.146** | 1.182 | 1.228 | 1.276 | 1.253 | 1.209 | 1.520 | 1.233 | 1.180 | 1.252 | 1.227 |
| | OWA | **0.873** | 0.890 | 0.922 | 0.950 | 0.938 | 0.911 | 1.106 | 0.921 | 0.893 | 0.936 | 0.923 |
| Month. | SMAPE | **12.410** | 12.710 | 12.995 | 13.416 | 13.258 | 13.079 | 13.514 | 12.959 | 13.059 | 13.089 | 13.513 |
| | MASE | **0.912** | 0.934 | 0.970 | 1.045 | 1.003 | 0.974 | 1.037 | 0.970 | 1.013 | 0.996 | 1.039 |
| | OWA | **0.859** | 0.880 | 0.910 | 0.957 | 0.931 | 0.911 | 0.929 | 0.905 | 0.929 | 0.922 | 0.957 |
| Others. | SMAPE | 4.721 | 4.843 | 4.805 | 4.973 | 6.124 | 4.780 | 6.709 | 4.952 | **4.711** | 6.599 | 6.913 |
| | MASE | 3.105 | 3.277 | 3.247 | 3.412 | 4.116 | 3.231 | 4.953 | 3.347 | **3.054** | 4.43 | 4.507 |
| | OWA | 0.986 | 1.026 | 1.017 | 1.053 | 1.259 | 1.012 | 1.487 | 1.049 | **0.977** | 1.393 | 1.438 |
| Avg. | SMAPE | **11.659** | 11.831 | 12.021 | 12.494 | 12.690 | 12.142 | 13.639 | 12.059 | 12.035 | 12.25 | 12.88 |
| | MASE | **1.557** | 1.585 | 1.612 | 1.731 | 1.808 | 1.631 | 2.095 | 1.623 | 1.625 | 1.698 | 1.836 |
| | OWA | **0.837** | 0.850 | 0.857 | 0.913 | 0.940 | 0.874 | 1.051 | 0.869 | 0.869 | 0.896 | 0.955 |

## F.3 TIME SERIES CLASSIFICATION

Table 10 summarizes the full results of time series classification. The baseline results are from existing works (Zhou et al., 2023a; Wu et al., 2022). From Table 10, we can observe that MSH-LLM achieves an average accuracy of 75.38%, surpassing all baselines including the best baseline FPT (74%) and TimesNet (73.6%).

Table 10: Full results of time series classification. We follow the protocol of existing work (Zhou et al., 2023a). The results are averaged from 10 subsets of UEA and higher values mean better performance. **Bolded**: Best, Underlined: Second best. # in the Transformers means the name of #former.

| Methods | LLM4TS | | Transformers | | | | | | | | | CNN | | MLP | | RNN | | Classical methods | |
|---|---|---|---|---|---|---|---|---|---|---|---|---|---|---|---|---|---|---|---|
| | MSH-LLM | FPT | Trans# | Re# | In# | Pyra# | Auto# | Station# | FED# | ETS# | Flow# | TimesNet | TCN | DLinear | LightTS. | LSTNet | LSSL | XGBoost | Rocket |
| EthanolConcentration | 36.2 | 34.2 | 32.7 | 31.9 | 31.6 | 30.8 | 31.6 | 32.7 | 31.2 | 28.1 | 33.8 | 35.7 | 28.9 | 32.6 | 29.7 | 39.9 | 31.1 | 43.7 | **45.2** |
| FaceDetection | **69.7** | 69.2 | 67.3 | 68.6 | 67 | 65.7 | 68.4 | 68 | 66 | 66.3 | 67.6 | 68.6 | 52.8 | 68 | 67.5 | 65.7 | 66.7 | 63.3 | 64.7 |
| Handwriting | 33.5 | 32.7 | 32 | 27.4 | 32.8 | 29.4 | 36.7 | 31.6 | 28 | 32.5 | 33.8 | 32.1 | 53.3 | 27 | 26.1 | 25.8 | 24.6 | 15.8 | **58.8** |
| Heartbeat | **80.9** | 77.2 | 76.1 | 77.1 | 80.5 | 75.6 | 74.6 | 73.7 | 73.7 | 71.2 | 77.6 | 78 | 75.6 | 75.1 | 75.1 | 77.1 | 72.7 | 73.2 | 75.6 |
| JapaneseVowels | 97.3 | 98.6 | 98.7 | 97.8 | 98.9 | 98.4 | 96.2 | **99.2** | 98.4 | 95.9 | 98.9 | 98.4 | 98.3 | 98.9 | 96.2 | 96.2 | 98.1 | 98.4 | 86.5 | 96.2 |
| PEMS-SF | 91.2 | 87.9 | 82.1 | 82.7 | 81.5 | 83.2 | 82.7 | 87.3 | 80.9 | 86 | 83.8 | 89.6 | 68.8 | 75.1 | 88.4 | 86.7 | 86.1 | **98.3** | 75.1 |
| SelfRegulationSCP1 | **93.5** | 93.2 | 92.2 | 90.4 | 90.1 | 88.1 | 84 | 89.4 | 88.7 | 89.6 | 92.5 | 91.8 | 84.6 | 87.3 | 89.8 | 84 | 90.8 | 84.6 | 90.8 |
| SelfRegulationSCP2 | **59.8** | 59.4 | 53.9 | 56.7 | 53.3 | 53.3 | 50.6 | 57.2 | 54.4 | 55 | 56.1 | 57.2 | 55.6 | 50.5 | 51.1 | 52.8 | 52.2 | 48.9 | 53.3 |
| SpokenArabicDigits | 99 | 99.2 | 98.4 | 97 | **100** | 99.6 | **100** | **100** | **100** | **100** | 98.8 | 99 | 95.6 | 81.4 | **100** | **100** | **100** | 69.6 | 71.2 |
| UWaveGestureLibrary | 92.7 | 88.1 | 85.6 | 85.6 | 85.6 | 83.4 | 85.9 | 87.5 | 85.3 | 85 | 86.6 | 85.3 | 88.4 | 82.1 | 80.3 | 87.8 | 85.9 | 75.9 | **94.4** |
| Average | **75.38** | 74 | 71.9 | 71.5 | 72.1 | 70.8 | 71.1 | 72.7 | 70.7 | 71 | 73 | 73.6 | 70.3 | 67.5 | 70.4 | 71.8 | 70.9 | 66 | 72.5 |

## F.4 FEW-SHOT LEARNING

We follow the same protocol of existing work (Pan et al., 2024). Table 11 and Table 12 summarize the results of few-shot learning under 10% training data. In the scope of 10% few-shot learning, MSH-LLM achieves SOTA results in almost all cases. Specifically, MSH-LLM achieves an average error reduction of 7.32% and 3.95% compared to LLM4TS methods (i.e., $S^2$IP-LLM and Time-LLM) in MSE and MAE, respectively.

Table 13 summarizes the average results and full results of few-shot learning under 5% training data. We can observe that MSH-LLM still achieves SOTA results even with fewer training data. Specifically, MSH-LLM achieves an average error reduction of 10.47% and 6.74% compared to LLM4TS methods (i.e., $S^2$IP-LLM and Time-LLM) in MSE and MAE, respectively.

Table 11: Few-shot learning results under 10% training data setting. Results are averaged from all forecasting lengths. **Bolded**: Best, Underlined: Second best. Full results are given in Table12.

| Methods | MSH-LLM (Ours) | $S^2$IP-LLM (ICML 2024) | Time-LLM (ICLR 2024) | FPT (NeurIPS 2023) | iTransformer (ICLR 2024) | PatchTST (ICLR 2023) | TimesNet (ICLR 2023) | FEDformer (ICML 2022) | NSFormer (NeurIPS 2022) | ETSformer (arXiv 2022) | Autoformer (NeurIPS 2021) |
|---|---|---|---|---|---|---|---|---|---|---|---|
| Metric | MSE MAE | MSE MAE | MSE MAE | MSE MAE | MSE MAE | MSE MAE | MSE MAE | MSE MAE | MSE MAE | MSE MAE | MSE MAE |
| Weather | **0.230 0.267** | 0.233 0.272 | 0.237 0.275 | 0.238 0.275 | 0.308 0.338 | 0.242 0.279 | 0.279 0.301 | 0.284 0.324 | 0.318 0.323 | 0.318 0.360 | 0.300 0.342 |
| Electricity | **0.167 0.260** | 0.175 0.271 | 0.177 0.273 | 0.176 0.269 | 0.196 0.293 | 0.180 0.273 | 0.323 0.392 | 0.346 0.427 | 0.444 0.480 | 0.660 0.617 | 0.431 0.478 |
| Traffic | **0.423 0.296** | 0.427 0.307 | 0.429 0.307 | 0.440 0.310 | 0.495 0.361 | 0.430 0.305 | 0.951 0.535 | 0.663 0.425 | 1.453 0.815 | 1.914 0.936 | 0.749 0.446 |
| ETTh1 | **0.563 0.514** | 0.593 0.529 | 0.785 0.553 | 0.785 0.553 | 0.910 0.860 | 0.590 0.525 | 0.633 0.542 | 0.869 0.628 | 0.639 0.561 | 0.915 0.639 | 1.180 0.834 | 0.702 0.596 |
| ETTh2 | **0.392** 0.423 | 0.419 0.439 | 0.424 0.441 | 0.397 **0.421** | 0.489 0.483 | 0.415 0.431 | 0.479 0.465 | 0.466 0.475 | 0.462 0.455 | 0.894 0.713 | 0.488 0.499 |
| ETTm1 | **0.403 0.424** | 0.455 0.435 | 0.487 0.461 | 0.464 0.441 | 0.728 0.565 | 0.501 0.466 | 0.677 0.537 | 0.722 0.605 | 0.797 0.578 | 0.980 0.714 | 0.802 0.628 |
| ETTm2 | **0.280 0.327** | 0.284 0.332 | 0.305 0.344 | 0.293 0.335 | 0.336 0.373 | 0.296 0.343 | 0.320 0.353 | 0.463 0.488 | 0.332 0.366 | 0.447 0.487 | 1.342 0.930 |

## F.5 ZERO-SHOT LEARNING

To evaluate the generalization ability of MSH-LLM, we perform zero-shot transfer experiments on M3 and M4 datasets. For M4 → M3 transfer, we evaluate the models trained on M4 directly on M3. Specifically, for the Yearly, Quarterly, and Monthly subsets of M3, we assign the corresponding M4-trained models based on their sampling frequencies. For the subsets of M3 Others, the M4 Quarterly model is selected to ensure consistency in forecasting horizons. For M3 → M4 transfer, a similar strategy is adopted: For the Yearly, Quarterly, and Monthly subsets of M4, the corresponding models trained on M3 datasets are used. To handle M4 Weekly, Daily, and Hourly subsets, we conduct inference using the M3 Monthly model, which is consistent with the experimental settings of existing methods (Zhou et al., 2023a; Liu et al., 2024).

The comprehensive zero-shot learning results are detailed in Table 14. Remarkably, MSH-LLM establishes a new state-of-the-art performance, consistently outperforming all competitive baselines

Table 12: Full results of few-shot learning under 10% training data. The input length is set to 512, and the forecasting lengths are set to 96, 192, 336, and 720. Smaller values correspond to better performance. **Bolded**: Best, Underlined: Second best.

| Methods | | MSH-LLM (Ours) | S²IP-LLM (ICML 2024) | Time-LLM (ICLR 2024) | FPT (NeurIPS 2023) | iTransformer (ICLR 2024) | PatchTST (ICLR 2024) | TimesNet (ICLR 2023) | FEDformer (ICML 2022) | NSFormer (NeurIPS 2022) | ETSformer (arXiv 2022) | Autoformer (NeurIPS 2021) |
|---|---|---|---|---|---|---|---|---|---|---|---|---|
| Metric | | MSE MAE | MSE MAE | MSE MAE | MSE MAE | MSE MAE | MSE MAE | MSE MAE | MSE MAE | MSE MAE | MSE MAE | MSE MAE |
| Weather | 96 | **0.152 0.208** | 0.159 0.210 | 0.160 0.213 | 0.163 0.215 | 0.253 0.307 | 0.165 0.215 | 0.184 0.230 | 0.188 0.253 | 0.192 0.234 | 0.199 0.272 | 0.221 0.297 |
| | 192 | 0.206 **0.248** | **0.200** 0.251 | 0.204 0.254 | 0.210 0.254 | 0.292 0.328 | 0.210 0.257 | 0.245 0.283 | 0.250 0.304 | 0.269 0.295 | 0.279 0.332 | 0.270 0.322 |
| | 336 | **0.252 0.286** | 0.257 0.293 | 0.255 0.291 | 0.256 0.292 | 0.322 0.346 | 0.259 0.297 | 0.305 0.321 | 0.312 0.346 | 0.370 0.357 | 0.356 0.386 | 0.320 0.351 |
| | 720 | **0.311 0.326** | 0.317 0.335 | 0.329 0.345 | 0.321 0.339 | 0.365 0.374 | 0.332 0.346 | 0.381 0.371 | 0.387 0.393 | 0.441 0.405 | 0.437 0.448 | 0.390 0.396 |
| | Avg | **0.230 0.267** | 0.233 0.272 | 0.237 0.275 | 0.238 0.275 | 0.308 0.338 | 0.242 0.279 | 0.279 0.301 | 0.284 0.324 | 0.318 0.323 | 0.318 0.360 | 0.300 0.342 |
| Electricity | 96 | **0.139 0.235** | 0.143 0.243 | 0.137 0.240 | 0.139 0.237 | 0.154 0.257 | 0.140 0.238 | 0.299 0.373 | 0.231 0.323 | 0.420 0.466 | 0.599 0.587 | 0.261 0.348 |
| | 192 | **0.153 0.248** | 0.159 0.258 | 0.159 0.258 | 0.156 0.252 | 0.171 0.272 | 0.160 0.255 | 0.305 0.379 | 0.261 0.356 | 0.411 0.459 | 0.620 0.598 | 0.338 0.406 |
| | 336 | **0.169 0.263** | 0.170 0.269 | 0.181 0.278 | 0.175 0.270 | 0.196 0.295 | 0.180 0.276 | 0.319 0.391 | 0.360 0.445 | 0.434 0.473 | 0.662 0.619 | 0.410 0.474 |
| | 720 | **0.207 0.295** | 0.230 0.315 | 0.232 0.317 | 0.233 0.317 | 0.263 0.348 | 0.241 0.323 | 0.369 0.426 | 0.530 0.585 | 0.510 0.521 | 0.757 0.664 | 0.715 0.685 |
| | Avg | **0.167 0.260** | 0.175 0.271 | 0.177 0.273 | 0.176 0.269 | 0.196 0.293 | 0.180 0.273 | 0.323 0.392 | 0.346 0.427 | 0.444 0.480 | 0.660 0.617 | 0.431 0.478 |
| Traffic | 96 | 0.405 **0.286** | **0.403** 0.293 | 0.406 0.295 | 0.414 0.297 | 0.448 0.329 | 0.403 0.289 | 0.719 0.416 | 0.639 0.400 | 1.412 0.802 | 1.643 0.855 | 0.672 0.405 |
| | 192 | 0.415 **0.286** | **0.412** 0.295 | 0.416 0.300 | 0.426 0.301 | 0.487 0.360 | 0.415 0.296 | 0.748 0.428 | 0.637 0.416 | 1.419 0.806 | 1.641 0.854 | 0.727 0.424 |
| | 336 | **0.417 0.293** | 0.427 0.316 | 0.430 0.309 | 0.434 0.312 | 0.514 0.372 | 0.426 0.304 | 0.853 0.471 | 0.655 0.427 | 1.443 0.815 | 1.711 0.878 | 0.749 0.454 |
| | 720 | **0.453 0.319** | 0.469 0.325 | 0.467 0.324 | 0.487 0.337 | 0.532 0.383 | 0.474 0.331 | 1.485 0.825 | 0.722 0.456 | 1.539 0.837 | 2.660 1.157 | 0.847 0.499 |
| | Avg | **0.423 0.296** | 0.427 0.307 | 0.429 0.307 | 0.440 0.310 | 0.495 0.361 | 0.430 0.305 | 0.951 0.535 | 0.663 0.425 | 1.453 0.815 | 1.914 0.936 | 0.749 0.446 |
| ETTh1 | 96 | 0.460 0.450 | 0.481 0.474 | 0.720 0.533 | **0.458 0.456** | 0.790 0.586 | 0.516 0.485 | 0.861 0.628 | 0.512 0.499 | 0.918 0.639 | 1.112 0.806 | 0.613 0.552 |
| | 192 | **0.516 0.488** | 0.518 0.491 | 0.747 0.545 | 0.570 0.516 | 0.837 0.609 | 0.598 0.524 | 0.797 0.593 | 0.624 0.555 | 0.915 0.629 | 1.155 0.823 | 0.722 0.598 |
| | 336 | **0.594 0.537** | 0.664 0.570 | 0.793 0.551 | 0.608 0.535 | 0.780 0.575 | 0.657 0.550 | 0.941 0.648 | 0.691 0.574 | 0.939 0.644 | 1.179 0.832 | 0.750 0.619 |
| | 720 | **0.680 0.581** | 0.711 0.584 | 0.880 0.584 | 0.725 0.591 | 1.234 0.811 | 0.762 0.610 | 0.877 0.641 | 0.728 0.614 | 0.887 0.645 | 1.273 0.874 | 0.721 0.616 |
| | Avg | **0.563 0.514** | 0.593 0.529 | 0.785 0.553 | 0.590 0.525 | 0.910 0.860 | 0.633 0.542 | 0.869 0.628 | 0.639 0.561 | 0.915 0.639 | 1.180 0.834 | 0.702 0.596 |
| ETTh2 | 96 | **0.331 0.366** | 0.354 0.400 | 0.334 0.381 | 0.331 0.374 | 0.404 0.435 | 0.353 0.389 | 0.378 0.409 | 0.382 0.416 | 0.389 0.411 | 0.678 0.619 | 0.413 0.451 |
| | 192 | **0.374 0.414** | 0.401 0.423 | 0.430 0.438 | 0.402 0.411 | 0.470 0.474 | 0.403 0.414 | 0.490 0.467 | 0.478 0.474 | 0.473 0.455 | 0.785 0.666 | 0.474 0.477 |
| | 336 | **0.396 0.432** | 0.442 0.450 | 0.449 0.458 | 0.406 0.433 | 0.489 0.485 | 0.426 0.441 | 0.537 0.494 | 0.504 0.501 | 0.477 0.472 | 0.839 0.694 | 0.547 0.543 |
| | 720 | **0.465 0.478** | 0.480 0.486 | 0.485 0.490 | 0.449 0.464 | 0.593 0.538 | 0.477 0.480 | 0.510 0.491 | 0.499 0.509 | 0.507 0.480 | 1.273 0.874 | 0.516 0.523 |
| | Avg | **0.392 0.423** | 0.419 0.439 | 0.424 0.441 | 0.397 0.421 | 0.489 0.483 | 0.415 0.431 | 0.479 0.465 | 0.466 0.475 | 0.462 0.455 | 0.894 0.713 | 0.488 0.499 |
| ETTm1 | 96 | **0.349 0.383** | 0.388 0.401 | 0.412 0.422 | 0.390 0.404 | 0.709 0.556 | 0.410 0.419 | 0.583 0.501 | 0.578 0.518 | 0.761 0.568 | 0.911 0.688 | 0.774 0.614 |
| | 192 | **0.377 0.410** | 0.422 0.421 | 0.447 0.438 | 0.429 0.423 | 0.717 0.548 | 0.437 0.434 | 0.630 0.528 | 0.617 0.546 | 0.781 0.574 | 0.955 0.703 | 0.754 0.592 |
| | 336 | **0.405 0.434** | 0.456 0.430 | 0.497 0.465 | 0.469 0.439 | 0.735 0.575 | 0.476 0.454 | 0.725 0.568 | 0.998 0.775 | 0.803 0.587 | 0.991 0.719 | 0.869 0.677 |
| | 720 | **0.482 0.468** | 0.554 0.490 | 0.594 0.521 | 0.569 0.498 | 0.752 0.584 | 0.681 0.556 | 0.769 0.549 | 0.693 0.579 | 0.844 0.581 | 1.062 0.747 | 0.810 0.630 |
| | Avg | **0.403 0.424** | 0.455 0.435 | 0.487 0.461 | 0.464 0.441 | 0.728 0.565 | 0.501 0.466 | 0.677 0.537 | 0.722 0.605 | 0.797 0.578 | 0.980 0.714 | 0.802 0.628 |
| ETTm2 | 96 | **0.178 0.261** | 0.192 0.274 | 0.224 0.296 | 0.192 0.285 | 0.245 0.322 | 0.191 0.274 | 0.212 0.285 | 0.291 0.399 | 0.229 0.308 | 0.331 0.430 | 0.352 0.454 |
| | 192 | **0.238 0.304** | 0.246 0.313 | 0.260 0.317 | 0.251 0.309 | 0.274 0.338 | 0.252 0.317 | 0.270 0.323 | 0.307 0.379 | 0.291 0.343 | 0.400 0.464 | 0.694 0.691 |
| | 336 | 0.299 **0.341** | 0.301 0.340 | 0.312 0.349 | 0.307 0.346 | 0.361 0.394 | 0.306 0.353 | 0.323 0.353 | 0.543 0.559 | 0.348 0.376 | 0.469 0.498 | 2.408 1.407 |
| | 720 | 0.403 0.401 | **0.400** 0.403 | 0.424 0.416 | 0.426 0.417 | 0.467 0.442 | 0.433 0.427 | 0.474 0.449 | 0.712 0.614 | 0.461 0.438 | 0.589 0.557 | 1.913 1.166 |
| | Avg | **0.280 0.327** | 0.284 0.332 | 0.305 0.344 | 0.293 0.335 | 0.336 0.373 | 0.296 0.343 | 0.320 0.353 | 0.463 0.488 | 0.332 0.366 | 0.447 0.487 | 1.342 0.930 |

Table 13: Full results of few-shot learning under 5% training data. The input length is set to 512, and the forecasting lengths are set to 96, 192, 336, and 720. '- -' indicates 5% training data is insufficient to constitute a training set. **Bolded**: Best, Underlined: Second best.

| Methods | | MSH-LLM (Ours) | S²IP-LLM (ICML 2024) | Time-LLM (ICLR 2024) | FPT (NeurIPS 2023) | iTransformer (ICLR 2024) | PatchTST (ICLR 2024) | TimesNet (ICLR 2023) | FEDformer (ICML 2022) | NSFormer (NeurIPS 2022) | ETSformer (arXiv 2022) | Autoformer (NeurIPS 2021) |
|---|---|---|---|---|---|---|---|---|---|---|---|---|
| Metric | | MSE MAE | MSE MAE | MSE MAE | MSE MAE | MSE MAE | MSE MAE | MSE MAE | MSE MAE | MSE MAE | MSE MAE | MSE MAE |
| Weather | 96 | **0.170 0.214** | 0.175 0.228 | 0.176 0.230 | 0.175 0.230 | 0.264 0.307 | 0.171 0.224 | 0.207 0.253 | 0.229 0.309 | 0.215 0.252 | 0.218 0.295 | 0.227 0.299 |
| | 192 | **0.213 0.253** | 0.225 0.271 | 0.226 0.275 | 0.227 0.276 | 0.284 0.326 | 0.230 0.271 | 0.272 0.307 | 0.265 0.317 | 0.290 0.307 | 0.294 0.331 | 0.278 0.333 |
| | 336 | **0.259 0.289** | 0.282 0.321 | 0.292 0.325 | 0.286 0.322 | 0.323 0.349 | 0.294 0.326 | 0.313 0.328 | 0.353 0.392 | 0.353 0.348 | 0.359 0.398 | 0.351 0.393 |
| | 720 | **0.346 0.367** | 0.361 0.371 | 0.364 0.375 | 0.366 0.379 | 0.366 0.375 | 0.384 0.387 | 0.400 0.385 | 0.391 0.394 | 0.452 0.407 | 0.461 0.461 | 0.387 0.389 |
| | Avg | **0.247 0.281** | 0.260 0.297 | 0.264 0.301 | 0.263 0.301 | 0.309 0.339 | 0.269 0.303 | 0.298 0.318 | 0.309 0.353 | 0.327 0.328 | 0.333 0.371 | 0.310 0.353 |
| Electricity | 96 | 0.144 0.243 | 0.148 0.248 | 0.148 0.248 | **0.143 0.241** | 0.162 0.264 | 0.145 0.244 | 0.315 0.389 | 0.235 0.322 | 0.484 0.518 | 0.697 0.638 | 0.297 0.367 |
| | 192 | **0.158 0.255** | 0.159 0.255 | 0.160 0.251 | 0.159 0.255 | 0.180 0.278 | 0.163 0.260 | 0.318 0.396 | 0.247 0.341 | 0.501 0.531 | 0.718 0.648 | 0.308 0.375 |
| | 336 | 0.177 0.272 | 0.175 0.271 | 0.183 0.282 | 0.179 0.274 | 0.207 0.305 | 0.183 0.281 | 0.340 0.415 | 0.267 0.356 | 0.574 0.578 | 0.758 0.667 | 0.354 0.411 |
| | 720 | **0.217 0.304** | 0.235 0.326 | 0.236 0.329 | 0.233 0.323 | 0.258 0.339 | 0.233 0.323 | 0.635 0.613 | 0.318 0.394 | 0.952 0.786 | 1.028 0.788 | 0.426 0.466 |
| | Avg | **0.174 0.269** | 0.179 0.275 | 0.181 0.279 | 0.178 0.273 | 0.201 0.296 | 0.181 0.277 | 0.402 0.453 | 0.266 0.353 | 0.627 0.603 | 0.800 0.685 | 0.346 0.404 |
| Traffic | 96 | **0.404 0.273** | 0.410 0.288 | 0.414 0.293 | 0.419 0.298 | 0.431 0.312 | 0.404 0.286 | 0.854 0.492 | 0.670 0.421 | 1.468 0.821 | 1.643 0.855 | 0.795 0.481 |
| | 192 | **0.405 0.291** | 0.416 0.298 | 0.419 0.300 | 0.434 0.305 | 0.456 0.326 | 0.412 0.294 | 0.894 0.517 | 0.653 0.405 | 1.509 0.838 | 1.856 0.928 | 0.837 0.503 |
| | 336 | **0.428 0.312** | 0.435 0.313 | 0.438 0.315 | 0.449 0.313 | 0.465 0.334 | 0.439 0.310 | 0.853 0.471 | 0.707 0.445 | 1.602 0.860 | 2.080 0.999 | 0.867 0.523 |
| | 720 | - - | - - | - - | - - | - - | - - | - - | - - | - - | - - | - - |
| | Avg | **0.413 0.292** | 0.420 0.299 | 0.423 0.302 | 0.434 0.305 | 0.450 0.324 | 0.418 0.296 | 0.867 0.493 | 0.676 0.423 | 1.526 0.839 | 1.859 0.927 | 0.833 0.502 |
| ETTh1 | 96 | **0.489 0.475** | 0.500 0.493 | 0.732 0.556 | 0.543 0.506 | 0.808 0.610 | 0.557 0.519 | 0.892 0.625 | 0.593 0.529 | 0.952 0.650 | 1.169 0.832 | 0.681 0.570 |
| | 192 | 0.658 0.535 | 0.690 0.539 | 0.872 0.604 | 0.748 0.580 | 0.928 0.658 | 0.711 0.570 | 0.940 0.665 | 0.652 0.563 | 0.943 0.645 | 1.221 0.853 | 0.725 0.602 |
| | 336 | 0.738 0.600 | 0.761 0.620 | 1.071 0.721 | 0.754 0.595 | 1.475 0.861 | 0.816 0.619 | 0.945 0.653 | 0.731 0.594 | 0.935 0.644 | 1.179 0.832 | 0.761 0.624 |
| | 720 | - - | - - | - - | - - | - - | - - | - - | - - | - - | - - | - - |
| | Avg | **0.628 0.537** | 0.650 0.550 | 0.891 0.627 | 0.681 0.560 | 1.070 0.710 | 0.694 0.569 | 0.925 0.647 | 0.658 0.562 | 0.943 0.646 | 1.189 0.839 | 0.722 0.598 |
| ETTh2 | 96 | **0.342 0.389** | 0.363 0.409 | 0.399 0.420 | 0.376 0.421 | 0.397 0.427 | 0.401 0.421 | 0.409 0.420 | 0.390 0.424 | 0.408 0.423 | 0.678 0.619 | 0.428 0.468 |
| | 192 | 0.375 0.412 | 0.375 0.411 | 0.487 0.479 | 0.418 0.441 | 0.438 0.445 | 0.452 0.455 | 0.483 0.464 | 0.457 0.465 | 0.497 0.468 | 0.845 0.697 | 0.496 0.504 |
| | 336 | **0.401 0.419** | 0.403 0.421 | 0.858 0.660 | 0.408 0.439 | 0.631 0.553 | 0.464 0.469 | 0.499 0.479 | 0.477 0.483 | 0.507 0.481 | 0.905 0.727 | 0.486 0.496 |
| | 720 | - - | - - | - - | - - | - - | - - | - - | - - | - - | - - | - - |
| | Avg | **0.373 0.407** | 0.380 0.413 | 0.581 0.519 | 0.400 0.433 | 0.488 0.475 | 0.827 0.615 | 0.439 0.448 | 0.463 0.454 | 0.470 0.489 | 0.809 0.681 | 0.441 0.457 |
| ETTm1 | 96 | **0.328 0.365** | 0.357 0.390 | 0.422 0.424 | 0.386 0.405 | 0.589 0.510 | 0.399 0.414 | 0.606 0.518 | 0.628 0.544 | 0.823 0.587 | 1.031 0.747 | 0.726 0.578 |
| | 192 | **0.353 0.395** | 0.432 0.434 | 0.448 0.440 | 0.440 0.438 | 0.703 0.565 | 0.441 0.436 | 0.681 0.539 | 0.666 0.566 | 0.844 0.591 | 1.087 0.766 | 0.750 0.591 |
| | 336 | **0.394 0.412** | 0.440 0.442 | 0.519 0.482 | 0.485 0.459 | 0.898 0.641 | 0.499 0.467 | 0.786 0.597 | 0.807 0.628 | 0.870 0.603 | 1.138 0.787 | 0.851 0.659 |
| | 720 | **0.518 0.483** | 0.455 0.446 | 0.593 0.521 | 0.577 0.499 | 0.948 0.671 | 0.767 0.587 | 0.796 0.593 | 0.822 0.633 | 0.893 0.611 | 1.245 0.831 | 0.857 0.655 |
| | Avg | **0.398 0.414** | 0.455 0.446 | 0.524 0.479 | 0.472 0.450 | 0.784 0.596 | 0.526 0.476 | 0.717 0.561 | 0.730 0.592 | 0.857 0.598 | 1.125 0.782 | 0.796 0.620 |
| ETTm2 | 96 | **0.179 0.264** | 0.197 0.278 | 0.225 0.300 | 0.199 0.280 | 0.265 0.339 | 0.206 0.288 | 0.220 0.299 | 0.229 0.320 | 0.238 0.316 | 0.404 0.485 | 0.232 0.322 |
| | 192 | **0.242 0.309** | 0.254 0.322 | 0.275 0.334 | 0.256 0.316 | 0.310 0.362 | 0.264 0.324 | 0.311 0.361 | 0.394 0.361 | 0.298 0.349 | 0.479 0.521 | 0.291 0.357 |
| | 336 | **0.300 0.344** | 0.315 0.350 | 0.339 0.371 | 0.318 0.353 | 0.373 0.399 | 0.334 0.367 | 0.338 0.366 | 0.378 0.427 | 0.353 0.380 | 0.552 0.555 | 0.478 0.517 |
| | 720 | **0.411 0.414** | 0.421 0.421 | 0.464 0.441 | 0.460 0.436 | 0.478 0.454 | 0.454 0.432 | 0.509 0.465 | 0.523 0.510 | 0.475 0.445 | 0.701 0.627 | 0.553 0.538 |
| | Avg | **0.283 0.333** | 0.296 0.342 | 0.325 0.361 | 0.308 0.346 | 0.356 0.388 | 0.314 0.352 | 0.344 0.372 | 0.381 0.404 | 0.341 0.372 | 0.534 0.547 | 0.388 0.433 |

Table 14: Full results of zero-shot learning. We adopt the same protocol of existing work (Pan et al., 2024). M4→M3 means training on M4 datasets and testing on M3 datasets, and vice versa. Lower SMAPE means better performance. **Bolded**: Best, Underlined: Second best.

| | Method | MSH-LLM (Ours) | AutoTimes (NeurIPS 2024) | FPT (NeurIPS 2023) | DLinear (AAAI 2023) | PatchTST (ICLR 2023) | TimesNet (ICLR 2023) | NSformer (NeurIPS 2022) | FEDformer (ICML 2022) | Informer (AAAI 2021) | Reformer (ICLR 2019) |
|---|---|---|---|---|---|---|---|---|---|---|---|
| M4 → M3 | Yearly | **15.650** | 15.710 | 16.420 | 17.430 | 15.990 | 18.750 | 17.050 | 16.000 | 19.700 | 16.030 |
| | Quarterly | **9.240** | 9.350 | 10.130 | 9.740 | 9.620 | 12.260 | 12.560 | 9.480 | 13.000 | 9.760 |
| | Monthly | **13.570** | 14.060 | 14.100 | 15.650 | 14.710 | 14.010 | 16.820 | 15.120 | 15.910 | 14.800 |
| | Others | 5.663 | 5.790 | | **4.810** | 6.810 | 9.440 | 6.880 | 8.130 | 8.940 | 13.030 | 7.530 |
| | Average | **12.469** | 12.750 | 13.060 | 14.030 | 13.390 | 14.170 | 15.290 | 13.530 | 15.820 | 13.370 |
| M3 → M4 | Yearly | **13.645** | 13.728 | 13.740 | 14.193 | 13.966 | 15.655 | 14.988 | 13.887 | 18.542 | 15.652 |
| | Quarterly | **10.703** | 10.742 | 10.787 | 18.856 | 10.929 | 11.877 | 11.686 | 11.513 | 16.907 | 11.051 |
| | Monthly | **14.489** | 14.558 | 14.630 | 14.765 | 14.664 | 16.165 | 16.098 | 18.154 | 23.454 | 15.604 |
| | Others | **6.132** | 6.259 | 7.081 | 9.194 | 7.087 | 6.863 | 6.977 | 7.529 | 7.348 | 7.001 |
| | Average | **12.968** | 13.036 | 13.125 | 15.337 | 13.228 | 14.553 | 14.327 | 15.047 | 19.047 | 14.092 |

across diverse transfer scenarios. On average, our approach achieves a substantial reduction in SMAPE error by over 10.23%. The performance gains are particularly pronounced in challenging tasks such as M4→M3 Others and M3→M4 Others, where the average error drops by 23.04% and 14.72%, respectively.

# G ABLATION STUDIES

**Multi-Scale Extraction (ME) Module.** To investigate the effectiveness of the ME module, we conduct an ablation study by carefully designing the following variant:

-w/o ME: Removing the multi-scale extraction module and only performs alignment between input time series and text prototypes.

Table 15: The results of different ME modules and hyperedging mechanisms on the ETTh1 dataset. The best results are **bolded**.

| Methods | -w/o ME | -w/o HM | -PM | MSH-LLM |
|---|---|---|---|---|
| Metirc | MSE MAE | MSE MAE | MSE MAE | MSE MAE |
| 96 | 0.412 0.400 | 0.693 0.560 | 0.380 0.392 | **0.360 0.388** |
| 192 | 0.413 0.412 | 0.751 0.513 | 0.405 0.424 | **0.398 0.411** |
| 336 | 0.421 0.436 | 0.756 0.596 | 0.423 0.443 | **0.415 0.432** |

The experimental results on the ETTh1 dataset are shown in Table 15. We can observe that MSH-LLM performs better than -w/o ME, showing the effectiveness of the ME module. The reason is that the ME module can provide richer representations than relying solely on single-scale alignment.

**Hyperedging Mechanism.** To investigate the effect of the hyperedging mechanism, we conduct an ablation study by carefully designing the following two variants:

-w/o HM: Removing the hyperedging mechanism and directly performing alignment between temporal features and text prototypes at different scales.

-PM: Replacing the hyperedging mechanism with the patching mechanism.

The experimental results on the ETTh1 dataset are shown in Table 15. We can observe that MSH-LLM performs better than -w/o HM and -PM, demonstrating the effectiveness of our hyperedging mechanism in enhancing the semantic information of time series semantic space. In addition, we can observe that -w/o HM achieves the worst performance, the reason is that the individual time point or temporal feature contains less semantic information, making it hard to align with the informative semantic space of natural language.

**Multi-Scale Text Prototypes Extraction.** To investigate the impact of different multi-scale text prototypes extraction, we conduct an ablation study by designing the following two variants:

R.1: Replacing word token embeddings based on pre-trained LLMs with word token embeddings generated from manually selected word and phrase descriptions (e.g., small, big, rapid increase, and steady decrease).

R.2: Replacing word token embeddings based on pre-trained LLMs with word token embeddings generated from randomly selected word and phrase descriptions (e.g., increase, happy, can, and white noise).

Table 16: The results of different multi-scale text prototypes extraction and CMA modules on the ETTh1 dataset. The best results are **bolded**.

| Methods | R.1 | R.2 | R.3 | P.1 | -ASO | MSH-LLM |
|---------|-----|-----|-----|-----|------|---------|
| Metric | MSE MAE | MSE MAE | MSE MAE | MSE MAE | MSE MAE | MSE MAE |
| 96 | 0.467 0.467 | 0.441 0.447 | 0.697 0.561 | 0.363 0.390 | 0.396 0.413 | **0.360 0.388** |
| 192 | 0.497 0.483 | 0.475 0.467 | 0.760 0.600 | 0.405 0.417 | 0.417 0.428 | **0.398 0.411** |
| 336 | 0.517 0.496 | 0.514 0.489 | 0.787 0.609 | 0.417 **0.424** | 0.433 0.442 | **0.415** 0.432 |

The experimental results on the ETTh1 dataset are shown in Table 16, from which we can observe that MSH-LLM performs better than R.1 and R.2 by a large margin, which indicates the effectiveness of our multi-scale text prototype extraction over approaches of manually selecting. In addition, it is notable that we initially assumed that aligning multi-scale temporal features with relevant natural language descriptions (e.g., small, big, rapid increase, and steady decrease) can offer better performance. However, the experimental results show that word token embeddings generated from randomly selected word and phrase descriptions achieve better performance than R.1. The reason is that the aligned word token embeddings may not be fully related to time series. Actually, LLMs can function as pattern recognition machines (Sun et al., 2024; Zhou et al., 2023a), and we believe the text prototypes matched by LLMs can better match temporal patterns, even if they may not be fully related to time series.

**CMA Module.** To investigate the effectiveness of the cross-modality alignment module, we conduct an ablation study by designing the following two variants:

R.3: Removing the CMA module and directly concatenating the hyperedge features with MoP before feeding them into LLMs to obtain the output representations.

P.1: Performing detailed cross-modality alignment across all scales.

The experimental results on the ETTh1 dataset are shown in Table 16, from which we can observe that MSH-LLM performs significantly better than R.3, showing the effectiveness of the CMA module. The reason is that the CMA module can help align the semantic space of natural language and that of time series. In addition, we can observe that MSH-LLM outperforms P.1 in most cases. This is because performing detailed alignment across all scales may introduce redundant information interference.

In addition, it has been shown that treating cross-modality alignment as an independent task (Li et al., 2023) can help the model focus more on the alignment objective and may potentially improve model performance. To investigate the impact of different cross-modality alignment strategies, we conduct ablation studies on the ETTh1 dataset by carefully designing the following variant:

-ASO: This approach treats cross-modality alignment as a standalone objective and employs a two-stage training strategy for time series analysis. The detailed design of the objective function is formulated as follows:

Specifically, for the given hyperedge feature $e_j^s$ and text prototypes $u_j^s$ at scale $s$, we first compute both the cosine similarity and the Euclidean distance between them. The cosine similarity can be formulated as follows:

$$\tau_{i,j} = \frac{e_i^s (e_j^s)^T}{\left\| e_i^s \right\|_2 \left\| e_j^s \right\|_2}, \tag{14}$$

where $||.||_2$ represents the L2 norm. The Euclidean distance can be defined as:

$$D_{i,j} = \left\| e_i^s - u_j^s \right\|_2 = \sqrt{\sum_{d=1}^{D} ((e_i^s)^d - (u_j^s)^d)^2} \tag{15}$$

Then, the loss function $L_{aso}^s$ at scale $s$ based on the correlation weight and Euclidean distance can be formulated as follows:

$$L_{aso}^s = \frac{1}{(M^s)^2} \sum_{i=1}^{M^s} \sum_{j=1}^{M^s} \left( \tau_{i,j} D_{i,j} + (1 - \tau_{i,j}) max(\gamma - D_{i,j}, 0) \right), \tag{16}$$

where $\gamma > 0$ denotes the threshold. Notably, when $\tau_{i,j} = 1$, indicating that $e_i^s$ and $u_k^s$ are deemed similar, the loss turns to $L_{aso} = \frac{1}{(M^s)^2} \sum_{i=1}^{M^s} \sum_{j=1}^{M^s} \tau_{i,j} D_{i,j}$, where the loss will increase if $D_{i,j}$

becomes large. Conversely, when $\alpha_{i,j} = 0$, meaning $e_i$ and $e_k$ are regarded as dissimilar, the loss turns to $L_{aso} = \frac{1}{(M^s)^2} \sum_{i=1}^{M^s} \sum_{j=1}^{M^s} (1 - \tau_{i,j}) max(\gamma - D_{i,j}, 0)$, where the loss will increase if $D_{i,j}$ falls below the threshold and turns smaller. Other cases lie between the above circumstances. The final loss function can be formulated as follows:

$$L = \sum\nolimits_{s=1}^{S} L_{aso}^s, \tag{17}$$

The experimental results are shown in Table 16. We can observe that MSH-LLM performs better than -ASO in most cases. We attribute the performance drop to the following two aspects: 1) Treating cross-modality alignment as a standalone objective, the model may lack supervision signals from the primary time series analysis task, thereby missing the potential synergy with the main task. 2) Unlike CV or NLP, time series datasets often contain limited training samples, which may result in insufficient generalization capability when cross-modality alignment is trained independently as a standalone objective. The experimental results show the effectiveness of our CMA module.

**LLM Backbones.** To investigate the effectiveness of LLM backbones for time series analysis, we conduct an ablation study by designing the following two variants:

-w/o LLM: Removing the LLM backbones and directly feeding the connected multi-scale temporal features into the linear mapping layer.

-LLM2Attn: Replacing the LLM backbones with a single multi-head attention layer.

Table 17: The results of LLM backbone variants on the ETTh1 dataset. The best results are **bolded**.

| Methods | -w/o LLM | -LLM2Attn | MSH-LLM |
|---|---|---|---|
| Metric | MSE MAE | MSE MAE | MSE MAE |
| 96 | 0.401 0.437 | 0.381 0.405 | **0.360 0.388** |
| 192 | 0.435 0.447 | 0.415 0.423 | **0.398 0.411** |
| 336 | 0.441 0.453 | 0.421 0.437 | **0.415 0.432** |

The experimental results on the ETTh1 dataset are shown in Table 17, from which we can observe that MSH-LLM performs better than -w/o LLM and -LLM2Attn, demonstrating the effectiveness of LLM backbones for time series analysis.

## H  VISUALIZATION

**Visualization of the Hyperedge Embeddings.** We perform qualitative analysis to investigate the training-time trajectories of the hyperedge embeddings. The t-SNE visualization results of hyperedge embeddings on the ETTh1 dataset are given in Figure 7. From Figure 7, we can discern the following tendencies: 1) As training progresses, hyperedge embeddings at different scales form distinct clusters. This indicates that MSH-LLM is able to distinguish and capture multi-scale temporal patterns. In addition, even within the same scale, different hyperedge embeddings reside in distinct clusters, indicating the ability of MSH-LLM in capturing diverse temporal patterns within the same scale. 2) From Figure 7(a) to Figure 7(c), we can observe that embeddings of large-scale hyperedges form distinct clusters earlier during training, while embeddings of small-scale hyperedges gradually separate from the large-scale clusters over time. This suggests that during the early stages of training, the model is more focused on capturing coarse-grained temporal patterns (e.g., weekly patterns), and later shifts its focus to learning finer-grained temporal patterns(e.g., hourly and daily patterns).

## I  MODEL ANALYSIS

### I.1  GENERALITY ANALYSIS ON DIFFERENT LLM BACKBONES.

For a fair comparison, following existing works (Liu et al., 2024; Pan et al., 2024), we use LLaMA-7B as the default LLM backbone. However, MSH-LLM is designed to enhance the general ability of LLMs to understand and process time series data, rather than being tailored to specific LLMs (e.g., LLaMA-7B). To evaluate the performance and generality of existing methods, we evaluate MSH-LLM

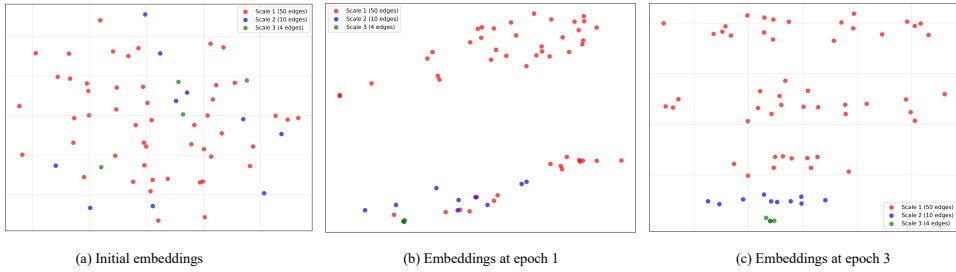

| (a) Initial embeddings | (b) Embeddings at epoch 1 | (c) Embeddings at epoch 3 |

Figure 7: The t-SNE visualization of hyperedge embeddings at different epochs.

with other baseline methods on more advanced LLMs. We adopt LLaMA-3.1-8B (Grattafiori et al., 2024) (-w L-8B), Qwen2.5-7B (Yang et al., 2024a) (-w Q-7B), and DeepSeek-R1-Distill-LLaMA-8B (Guo et al., 2025) (-w D-8B) for comparison. The experimental results on the ETTh1 dataset with input length T=512 and output length H=96 are presented in Table 18.

Table 18: The results of different LLM backbones on the ETTh1 dataset. The best results are **bolded**.

| Methods | -w L-8B | -w Q-7B | -w D-8B | LLaMA-7B (Default) |
|---------|---------|---------|---------|--------------------|
| Metric | MSE MAE | MSE MAE | MSE MAE | MSE MAE |
| $S^2$IP-LLM | 0.350 0.393 | 0.364 0.395 | 0.362 0.395 | 0.366 0.396 |
| Time-LLM | 0.378 0.403 | 0.379 0.413 | 0.378 0.408 | 0.383 0.410 |
| MSH-LLM | **0.350 0.377** | **0.352 0.383** | **0.348 0.365** | **0.360 0.388** |

From Table 18, we can observe that existing LLM4TS methods (i.e., MSH-LLM $S^2$IP-LLM, and Time-LLM) achieve better performance on more advanced LLMs, demonstrating the significance of the choice of LLM backbones for time series analysis. In addition, we can observe that MSH-LLM shows a more significant improvement compared to other methods when using more advanced LLM backbones. This indicates the effectiveness of the framework design, rather than being merely influenced by the LLM backbones.

## I.2 ROBUSTNESS ANALYSIS

All experimental results reported in the main text and appendix are averaged over three runs with different random seeds. To evaluate the robustness of MSH-LLM to the choice of random seeds, we report the standard deviation of MSH-LLM under long-term time series forecasting settings. The experimental results are shown in Table 19 and 20. We can observe that the variances are considerably small, which indicates the robustness of MSH-LLM against the choice of random seeds.

Table 19: The standard deviation results of MSH-LLM on Weather, Electricity, and Traffic datasets. Results are averaged from three random seeds.

| Dataset | Weather | Electricity | Traffic |
|---------|---------|-------------|---------|
| Horizon | MSE MAE | MSE MAE | MSE MAE |
| 96 | 0.138±0.0005 0.187±0.0007 | 0.127±0.0012 0.231±0.0005 | 0.365±0.0000 0.270±0.0003 |
| 192 | 0.187±0.0010 0.230±0.0009 | 0.150±0.0006 0.242±0.0003 | 0.372±0.0005 0.281±0.0002 |
| 336 | 0.237±0.0007 0.282±0.0003 | 0.162±0.0001 0.258±0.0000 | 0.385±0.0000 0.279±0.0003 |
| 720 | 0.305±0.0002 0.315±0.0001 | 0.198±0.0005 0.279±0.0003 | 0.402±0.0006 0.303±0.0009 |

In addition, it is notable that achieving significant performance improvement across all well-studied datasets is inherently challenging. To rule out the influence of experimental errors, instead of just showing the MSE and MAE results, we repeat all experiments 3 times and report the standard deviation and statistical significance level (T-test) of MSH-LLM and the second-best baseline (i.e., $S^2$IP-LLM). The experimental results are shown in Table 21.

From Table 21, we can observe that all the statistical significance reaches 95%, indicating that the performance improvements achieved by MSH-LLM are substantial and consistent across all datasets.

To evaluate the robustness of the proposed method, we compare MSH-LLM with baselines (i.e., $S^2$IP-LLM, Time-LLM, and FPT) across three challenging scenarios: forecasting with anomaly

Table 20: The standard deviation results of MSH-LLM on the ETT dataset. Results are averaged from three random seeds.

| Dataset | ETTh1 | | ETTh2 | | ETTm1 | | ETTm2 | |
|---|---|---|---|---|---|---|---|---|
| Horizon | MSE MAE | | MSE MAE | | MSE MAE | | MSE MAE | |
| 96 | 0.360±0.0007 0.388±0.0005 | | 0.273±0.0009 0.331±0.0004 | | 0.285±0.0031 0.340±0.0011 | | 0.161±0.0001 0.246±0.0005 | |
| 192 | 0.398±0.0014 0.411±0.0003 | | 0.335±0.0005 0.372±0.0003 | | 0.313±0.0016 0.358±0.0017 | | 0.218±0.0008 0.284±0.0003 | |
| 336 | 0.415±0.0010 0.432±0.0007 | | 0.363±0.0007 0.400±0.0000 | | 0.355±0.0068 0.377±0.0024 | | 0.271±0.0005 0.320±0.0003 | |
| 720 | 0.436±0.0003 0.447±0.0006 | | 0.396±0.0015 0.428±0.0009 | | 0.405±0.0121 0.410±0.0062 | | 0.358±0.0007 0.392±0.0004 | |

Table 21: The standard deviation and T-test results of MSH-LLM and the second-best baseline. Results are averaged from three random seeds.

| Dataset | MSH-LLM | S$^2$IP-LLM | Confidence Interval |
|---|---|---|---|
| Horizon | MSE MAE | MSE MAE | Percent |
| 96 | 0.217±0.0006 0.254±0.0005 | 0.223±0.0007 0.259±0.0005 | 99% |
| 192 | 0.159±0.0006 0.253±0.0003 | 0.163±0.0006 0.258±0.0005 | 99% |
| 336 | 0.381±0.0003 0.283±0.0004 | 0.406±0.0003 0.287±0.0004 | 99% |
| 720 | 0.334±0.0020 0.371±0.0010 | 0.338±0.0014 0.379±0.0010 | 95% |

injection, ultra-long forecasting, and forecasting with missing data. The corresponding results are presented below. Note that to quantify robustness, we compute the performance drop rate (PDR) as:

$$PDR = \frac{\Gamma - \hat{\Gamma}}{\Gamma} \tag{18}$$

where $\Gamma$ and $\hat{\Gamma}$ are forecasting results and forecasting results under challenging scenarios, respectively. Higher PDR values indicate lower robustness. The reported PDR is averaged across the MSE and MAE metrics to provide a comprehensive evaluation.

**Forecasting With Anomaly Injection.** We conduct experiments by injecting randomly generated anomalies in the training data. The anomaly rate varies from 10% to 20%. The experiments are conducted on the ETTh1 dataset with the input length set to 512 and output length set to 96. Table 1 summarizes the results of forecasting with anomaly injection.

Table 22: Forecasting results with anomaly injection on the ETTh1 dataset. The best results are **bolded**.

| Methods | MSH-LLM | S$^2$IP-LLM | Time-LLM | FPT |
|---|---|---|---|---|
| Metric | MSE MAE PDR | MSE MAE PDR | MSE MAE PDR | MSE MAE PDR |
| 0% | **0.360 0.388 /** | 0.366 0.396 / | 0.383 0.410 / | 0.379 0.402 / |
| 10% | **0.374 0.393 2.589** | 0.712 0.574 69.743 | 0.398 0.419 3.056 | 0.410 0.393 2.970 |
| 15% | **0.425 0.427 14.053** | 0.723 0.578 71.750 | 0.443 0.435 10.812 | 0.741 1.103 134.946 |
| 20% | **0.773 0.598 81.106** | 0.773 0.598 81.106 | 0.751 0.589 69.871 | 0.935 1.421 200.092 |

From Table 22, we can obtain the following tendencies: 1) MSH-LLM achieves the best performance in almost all cases, showing its superior ability in time series forecasting even under scenarios with anomaly injection. 2) Although the performance of all methods declines as the anomaly ratio increases, MSH-LLM exhibits a slower performance degradation compared to the other methods, demonstrating its robustness for forecasting with anomaly injection. 3) When the anomaly ratio reaches about 20%, the PDR value of MSH-LLM is greater than 20%, indicating that the robustness boundary of MSH-LLM is near 20% anomaly injection.

**Ultra-Long-Term Forecasting**. We conduct ultra-long-term time series forecasting by taking a fixed input length (T=512) to predict ultra-long horizons (H={1008, 1440, 1800}). Table 23 summarizes the results of ultra-long-term time series forecasting.

From Table 23, we can observe that MSH-LLM achieves SOTA results on almost all cases, showing the effectiveness of MSH-LLM for ultra-long-term time series forecasting. In addition, although all baselines suffer from performance drops when increasing forecasting horizons, MSH-LLM declines more gradually. The reason may be that the multi-scale hypergraph structure enhances the ability of LLMs in understanding and processing ultra-long-term time series.

Table 23: Ultra-long-term forecasting on the ETTh1 dataset. The best results are **bolded**.

| Methods | MSH-LLM | | S$^2$IP-LLM | | Time-LLM | | FPT | |
|---|---|---|---|---|---|---|---|---|
| Metric | MSE | MAE | MSE | MAE | MSE | MAE | MSE | MAE |
| 1008 | **0.463** | **0.498** | 0.543 | 0.520 | 0.478 | 0.475 | 0.527 | 0.576 |
| 1440 | **0.516** | **0.513** | 0.806 | 0.642 | 0.547 | 0.521 | 0.594 | 0.716 |
| 1800 | **0.648** | **0.557** | 0.940 | 0.725 | 0.683 | 0.5587 | 0.660 | 0.886 |

**Forecasting With Missing Data.** We conduct forecasting with missing data by randomly masking the training data. The experiments are conducted on the Electricity dataset with the input length set to 512 and output length set to 96. Table 24 summarizes the results of forecasting with missing data.

Table 24: Ultra-long-term forecasting on the ETTh1 dataset. The best results are **bolded**.

| Methods | MSH-LLM | | | S$^2$IP-LLM | | | Time-LLM | | | FPT | | |
|---|---|---|---|---|---|---|---|---|---|---|---|---|
| Metric | MSE | MAE | | MSE | MAE | | MSE | MAE | | MSE | MAE | |
| 0% | **0.360** | **0.388** | / | 0.366 | 0.396 | / | 0.383 | 0.410 | / | 0.379 | 0.402 | / |
| 5% | **0.368** | **0.3.93** | 3.511 | 0.385 | 0.403 | 3.479 | 0.392 | 0.416 | 1.907 | 0.392 | 0.431 | 5.307 |
| 10% | **0.409** | **0.421** | 11.058 | 0.432 | 0.447 | 15.456 | 0.451 | 0.449 | 13.633 | 0.478 | 0.483 | 22.704 |

From Table 24, we can obtain the following tendencies: 1) Existing LLM-based methods show little performance degradation with 5% missing data. The reason may be that LLM4TS methods can leverage transferable knowledge learned from large-scale corpora of sequences, thereby enhancing their abilities in understanding and reasoning time series. 2) MSH-LLM performs better than other LLM4TS methods, the reason is that the hyperedging mechanism can capture group-wise interactions, which increase the robustness of LLM in forecasting with missing data. 3) When the missing data ratio reaches about 10%, the PDR value of MSH-LLM is greater than 10%, indicating that the robustness boundary of MSH-LLM is near 10% missing data.

Table 25: Results compared with simple methods on the ETTh1 dataset. The best results are **bolded**.

| Methods | DHR-ARIMA | | Repeat | | PAtnn | | MSH-LLM | |
|---|---|---|---|---|---|---|---|---|
| Metric | MSE | MAE | MSE | MAE | MSE | MAE | MSE | MAE |
| 96 | 0.894 | 0.613 | 1.294 | 0.713 | 0.383 | 0.411 | **0.360** | **0.388** |
| 192 | 0.872 | 0.624 | 1.325 | 0.733 | 0.429 | 0.438 | **0.398** | **0.411** |
| 336 | 0.957 | 0.638 | 1.330 | 0.746 | 0.425 | 0.443 | **0.415** | **0.432** |

## I.3 BROADER BENCHMARK COMPARISON

**Comparison With Other Cross-Modality Alignment (CMA) Method.** It is known that TimeCMA (Liu et al., 2025a) also uses the CMA mechanism. We need to clarify that despite the shared nomenclature, the implementation and operational mechanisms of the cross-modality alignment (CMA) modules in MSH-LLM and TimeCMA are fundamentally different. Specifically, the CMA mechanism in TimeCMA operates primarily as a retrieval mechanism. Its goal is to query and extract time-series-related representations from a set of predefined, hand-crafted textual prompts. This process can be seen as a form of feature selection, where the most relevant linguistic cues are retrieved to augment the time series embeddings. In contrast, the CMA module in MSH-LLM operates primarily as an alignment and fusion mechanism. It is designed to align the multi-scale hyperedge features with multi-scale text prototypes generated from the token embeddings of LLMs. This process aims to align the modality between natural language and time series. In addition, we have included TimCMA for comparison. Note that due to its fixed prompt templates and restrictions on LLM selection, we failed to rerun TimeCMA under the unified settings. For a fair comparison, we reran MSH-LLM using the same settings as TimCMA. The experimental results on the ETTh1 dataset with the input length T=96 are shown in Table 26.

From Table 26, we can observe that MSH-LLM performs better than TimeCMA in almost all cases, demonstrating the effectiveness of CMA mechanism used in MSH-LLM.

**Comparison With Simple Methods.** Recent studies have questioned the effectiveness of previous LLM-based methods for time series analysis (Tan et al., 2024). Some studies even show that a

Table 26: Results compared with simple methods on the ETTh1 dataset. The best results are **bolded**.

| Methods | TimeCMA | | MSH-LLM | |
|---------|---------|-----|---------|-----|
| Metric | MSE | MAE | MSE | MAE |
| 96 | 0.373 | **0.391** | **0.362** | 0.393 |
| 192 | 0.427 | 0.421 | **0.417** | **0.416** |
| 336 | 0.458 | 0.448 | **0.420** | **0.423** |

simple attention layer or non-neural methods (Hewamalage et al., 2023) can achieve comparable performance. To further evaluate the performance of MSH-LLM against simple methods, we add three simple methods, i.e., PAtnn (Tan et al., 2024), DHR-ARIMA (Hewamalage et al., 2023), and Repeat (used in DLinear (Zeng et al., 2022)) for comparison. All experiments are run under unified settings. The experimental results on the ETTh1 dataset are shown in Table 25.

From Table 25, we can observe that MSH-LLM performs better than simple methods in most cases. Specifically, MSH-LLM reduces the MSE errors by 56.89%, 70.31%, and 5.20% compared to DHR-ARIMA, Repeat, and PAtnn, respectively. The experimental results demonstrate the effectiveness of MSH-LLM over simple methods.

Here, we attribute the limited effectiveness of previous LLM-based methods for time series analysis to three key factors: Firstly, the semantic spaces of natural language and time series are inherently different. Existing methods (e.g., FPT (Zhou et al., 2023a)) directly leverage off-the-shelf LLMs for time series analysis without proper alignment, making it difficult for LLMs to understand and process temporal features. Secondly, we found that some of these methods (e.g., CALF (Liu et al., 2025b) and FPT) do not even use prompts for LLMs, despite prompts being proven crucial for activating the reasoning capabilities of LLMs. The third and most important factor is that existing LLM-based methods directly segment the input time series into patches and feed them into LLMs. However, simple partitioning of patches may introduce noise interference and negatively impact the ability of LLMs to understand and process temporal information.

In contrast, our proposed method incorporate hyperedging mechanism, CMA module, and MoP mechanism, all of which are designed to better align LLMs for time series analysis. Ablation studies in Section 5.6 and Appendix G confirm that these components can enhance the ability of LLMs to understand and process temporal information. Experimental results in Appendix I.2 further validate the effectiveness of our method in both utilizing LLMs and addressing concerns about the performance ceiling of previous methods.

## J  PROOF

In our numerical experiments and visualization analysis, we find that different hyperedge representations capture distinct semantic information and can enhance the ability of LLMs in reasoning time series data. To further explore, we use the following theorem to characterize this behavior.

**Theorem 1 (Informal).** Consider the self-attention mechanism for the $l$-th query token. Assume that the input tokens $\boldsymbol{X}_i$ ($i = 1, 2, \ldots, n$) have a bounded mean $\mu$. Under mild conditions, with high probability, the output value token $\hat{\boldsymbol{X}}_i$ with high probability converges to $\mu W_i$ at a rate of $\mathcal{O}(n^{-1/2})$, where $W_i$ is the parameter matrix used to compute the value token.

This indicates that the self-attention mechanism used in LLMs can efficiently converge the output token representations to a stable mean (i.e., the representative semantic center). For time series analysis, if there are translation-invariant structures or patterns (e.g., periodicity and trend), the self-attention can help identify those invariant structures more effectively by comparing a given token with others. This phenomenon is especially important in few-shot forecasting or high-noise scenarios as it helps avoid overfitting to noise and improves generalization.

However, raw time series data suffer from two main limitations: 1) Individual time points contain limited semantic information, making it difficult to reflect structural patterns (e.g., periodicity and trend). 2) The raw sequence is often corrupted by noise, resulting in a low signal-to-noise ratio. To address these issues, we introduce multi-scale hypergraph structures, which adaptively connect multiple time points through learnable hyperedges at different scales. This method can enhance the

multi-scale semantic information of time series while reducing irrelevant information interference. It provides the self-attention mechanism in LLMs with more structured input, enabling self-attention to distinguish between temporal patterns and noise. As a result, the generalization and robustness of LLMs are improved.

We denote the $i$-th element of vector $\boldsymbol{X}$ as $x_i$, the element in the $i$-th row and $j$-th column of matrix $\mathbf{W}$ as $W_{ij}$, and the $j$-th row of matrix $\mathbf{W}$ as $\mathbf{W}_{j:}$. Furthermore, we denote the $i$-th hyperedge representation (token) of the input as $\mathbf{x}_i$, where $\mathbf{x}_i = \boldsymbol{X}_i$. Following existing work (Zhou et al., 2023a), before giving the formal statement of Theorem E.1, we first show the following three assumptions.

1. Each token $\mathbf{x}_i$ is a sub-Gaussian random vector with mean $\boldsymbol{\mu}_i$ and covariance matrix $(\sigma^2/d)\mathbf{I}$, for $i = 1, 2, \ldots, n$.

2. The mean vector $\boldsymbol{\mu}$ follows a discrete distribution over a finite set $\mathcal{V}$. Furthermore, there exist constants $0 < \nu_1$ and $0 < \nu_2 < \nu_4$ such that:

   a) $\|\boldsymbol{\mu}_i\| = \nu_1$,

   b) $\boldsymbol{\mu}_i^\top \mathbf{W}_Q \mathbf{W}_K^\top \boldsymbol{\mu}_i \in [\nu_2, \nu_4]$ for all $i$, and $|\boldsymbol{\mu}_i^\top \mathbf{W}_Q \mathbf{W}_K^\top \boldsymbol{\mu}_j| \leq \nu_2$ for all $\boldsymbol{\mu}_i \neq \boldsymbol{\mu}_j \in \mathcal{V}$.

3. The matrices $\mathbf{W}_V$ and $\mathbf{W}_Q \mathbf{W}_K^\top$ are element-wise bounded by $\nu_5$ and $\nu_6$, respectively. That is, $|[\mathbf{W}_V]_{ij}| \leq \nu_5$ and $|[\mathbf{W}_Q \mathbf{W}_K^\top]_{ij}| \leq \nu_6$ for all $i, j \in [d]$.

In the above assumptions, we ensure that for a given query hyperedge representation, the difference between the clustering center and noises are large enough to be distinguished. Then, we give the formal statement of Theorem 1 as follows:

**Theorem 2 (formal statement of Theorem 1).** Let each hyperedge representation $\mathbf{x}_i$ be a $\sigma$-subgaussian random vector with mean $\boldsymbol{\mu}_i$, and suppose all $n$ hyperedge representations share the same query cluster center. Under the aforementioned assumptions, if $\nu_1 > 3(\psi(\delta, d) + \nu_2 + \nu_4)$, then with probability at least $1 - 5\delta$, we have:

$$\left\| \frac{\sum_{i=1}^n \exp\left(\frac{1}{\sqrt{d}}\mathbf{x}_i \mathbf{W}_Q \mathbf{W}_K^\top \mathbf{x}_l^\top\right) \mathbf{x}_i \mathbf{W}_V}{\sum_{j=1}^n \exp\left(\frac{1}{\sqrt{d}}\mathbf{x}_j \mathbf{W}_Q \mathbf{W}_K^\top \mathbf{x}_l^\top\right)} - \boldsymbol{\mu}_l \mathbf{W}_V \right\|_\infty$$

$$\leq 4 \exp\left(\frac{\psi(\delta, d)}{\sqrt{d}}\right) \sigma\nu_5 \sqrt{\frac{2}{dn} \log\left(\frac{2d}{\delta}\right)}$$

$$+ 7 \left[\exp\left(\frac{\nu_2 - \nu_4 + \psi(\delta, d)}{\sqrt{d}}\right) - 1\right] \|\boldsymbol{\mu}_l \mathbf{W}_V\|_\infty,$$

where $\psi(\delta, d) = 2\sigma\nu_1\nu_6\sqrt{2\log\left(\frac{1}{\delta}\right)} + 2\sigma^2\nu_6 \log\left(\frac{d}{\delta}\right)$.

*Proof. See the proof of Lemma 2 in (Wang et al., 2022) with $k_1 = k = n$.*

## K  LIMITATIONS AND FUTURE WORK

In the future, we will extend our work in the following directions. Firstly, due to our CMA module performing multi-scale alignment in a fully learnable manner, it is interesting to introduce a constraint mechanism to further enhance the alignment between multi-scale temporal features and multi-scale text prototypes. Secondly, compared to natural language processing and computer vision, time series analysis has access to fewer datasets, which may limit the expressive power of the models. Therefore, in the future, we plan to compile larger datasets to validate the generalization capabilities of our models on more extensive data.

## L    USE OF LLMS

The authors use LLM solely as a general-purpose assistive tool for grammar and format refinement. LLM does not contribute to research ideation or experimental design. The authors take full responsibility for the content of this paper.

