# OpenReview forum: "Multi-Scale Hypergraph Meets LLMs: Aligning Large Language Models for Time Series Analysis"
_ICLR.cc/2026/Conference — ICLR 2026 Poster_

### Official Review · Reviewer_GFpP · 2025-10-27

[review text omitted: it was posted to a different submission]

---

> ### Author Response · Authors · 2025-11-23
>
> **Question 1**. It is better to train the multimodal alignment as a standalone objective (rather than only as an auxiliary loss) and report a dedicated set of results—ideally comparing standalone vs. auxiliary alignment settings.
>
> Thanks for your insightful comments and scientific rigor. We agree with you that treating multimodality alignment as an independent task **can help the model focus more on the alignment objective and may potentially improve model performance**. To investigate the impact of the multimodality alignment strategy, we conduct ablation studies on the ETTh1 dataset by carefully designing the following variant:
>
> -ASO: This approach treats cross-modality alignment as a standalone objective and employs a two-stage training strategy for time series analysis.
>
> The experimental results are shown in Table 1, and the detailed design of -ASO is provided in Appendix H of the $\underline{\text{revised paper}}$.
>
> Table 1.
>
> | Methods | -ASO        | MSH-LLM         |
> | ------- | ----------- | --------------- |
> | Metric  | MSE MAE     | MSE MAE         |
> | 96      | 0.396 0.413 | **0.360 0.388** |
> | 192     | 0.417 0.428 | **0.398 0.411** |
> | 336     | 0.433 0.442 | **0.415 0.432** |
>
> From Table 1, we can observe that MSH-LLM performs better than -ASO in most cases. We attribute the performance drop to the following two aspects: 1) Treating multimodality alignment as a standalone objective, the model may **lack supervision signals from the primary time series analysis task**, thereby missing the potential synergy with the main task. 2) Unlike CV or NLP, time series datasets often contain limited training samples, which may result in **insufficient generalization capability when cross-modality alignment is trained independently as a standalone objective**. However, the valuable feedback from the reviewer has inspired us to consider that cross-modality alignment **does not necessarily have to be treated as an auxiliary objective**. With sufficient training samples, **training the cross-modality alignment model independently and fine-tuning it on downstream tasks or adopting more effective training strategies may yield better representations** for time series analysis. Considering the scope of this paper, we would like to leave the exploration in future work.
>
> **Question 2**. It is better to provide a ablation study on the design of endogenous prompts (e.g., template variants, statistical descriptors, temporal granularity), to quantify their impact on the final performance.
>
> Thanks for your comments. To quantify the impact of endogenous prompts on the final performance, we have newly added ablation studies by carefully designing the following variants:
>
> -TV: It replaces the data-correlated prompts with the prompt template used in Time-LLM.
>
> -SD1: It incorporates more data statistics (e.g., trends, lags, means, and standard deviation) into the data-correlated prompts.
>
> -SD2: It selects a few key statistical metrics to include as data statistics in the data-correlated prompts.
>
> -TG: It incorporates different temporal granularity information into the data-correlated prompts.
>
> Table 2.
>
> | Methods | -TV         | -SD1            | -SD2            | -TG             | MSH-LLM         |
> | ------- | ----------- | --------------- | --------------- | --------------- | --------------- |
> | Metric  | MSE MAE     | MSE MAE         | MSE MAE         | MSE MAE         | MSE MAE         |
> | 96      | 0.363 0.389 | 0.360 0.389     | **0.358 0.388** | 0.359 0.390     | 0.360 **0.388** |
> | 192     | 0.398 0.413 | 0.400 **0.411** | **0.397** 0.413 | 0.398 0.413     | 0.398 **0.411** |
> | 336     | 0.417 0.435 | 0.415 0.433     | 0.414 **0.431** | **0.413** 0.432 | 0.415 0.432     |
>
> The experimental results are shown in Table 2, which has also been included in Appendix H of the $\underline{\text{revised paper}}$. From Table 2, we can obtain the following tendencies: 1) MSH-LLM performs better than -TV in most cases, **showing the effectiveness of our prompt template**. 2) -SD2 outperforms both -SD1 and MSH-LLM, suggesting that **more statistical features do not necessarily lead to better performance**, and **carefully selected statistical metrics may yield superior results**. 3) MSH-LLM achieves comparable performance to these variants. The reason is that **we design the mixture of prompts (MoP) mechanism**, which mitigates the impact of **relying on a single prompt or specific statistical features**. The experimental results demonstrate the robustness of our MoP mechanism.

---

> ### Author Response · Authors · 2025-11-23
>
> **Question 3**. The framework appears extensible to broader multimodal time-series tasks (e.g., classification and forecasting). It is better to evaluate on a few forecasting datasets, or more narrowly, on forecasting-based anomaly detection (early-warning) to assess transferability.
>
> Thanks for the comments. As a general time series analysis framework, MSH-LLM can be applied to different time-series tasks (e.g., classification and anomaly detection). The experimental results in Section 5.3 of the $\underline{\text{original paper}}$ show the **effectiveness of MSH-LLM for time series classification**. To further investigate the effectiveness of MSH-LLM in anomaly detection, we compare MSH-LLM with baselines on five commonly used datasets, including SMD, MSL, SMAP, SWaT, and PSM. The experimental settings follow those in existing works (e.g., FPT and TimesNet). The average F1-score is given in Table 3 and the full results are given in Appendix J.1 of the $\underline{\text{revised paper}}$.
>
> Table 3.
>
> | Methods | MSH-LLM   | FPT   | TimesNet | PatchTST | ETSformer | DLinear |
> | ------- | --------- | ----- | -------- | -------- | --------- | ------- |
> | Avg.    | **88.08** | 86.72 | 85.24    | 82.79    | 82.87     | 82.46   |
>
> As shown in Table 3, MSH-LLM achieves an average F1-score of 88.08%, **outperforming all baseline methods and highlighting its effectiveness in time series anomaly detection**. The experimental results indicate that MSH-LLM is capable of detecting infrequent anomalies in time series, which can be attributed to **the multi-scale hypergraph structure that enhances the reasoning capabilities of LLMs for modeling multi-scale temporal patterns**.

---

> ### Author Response · Authors · 2025-11-27
> **Request of Reviewer's attention and feedback**
>
> Dear Reviewer,
>
> Thank you again for your valuable review, which has greatly inspired us to improve our paper further. This is a kind reminder that only a few days remain in the discussion period. We kindly ask if our responses have addressed your concerns.
>
> The paper has been strengthened in line with your suggestions. Key enhancements include **incorporating ablation studies on cross-modality alignment**, **evaluating the design of endogenous prompts**, and **extending the comparison**​ of MSH-LLM with baselines on more time-series tasks.
>
> **Thank you again for your time, dedication, and constructive feedback**. We look forward to hearing your thoughts on our revisions.

---

### Official Review · Reviewer_GvBV · 2025-10-27

**Soundness:** 2
**Presentation:** 2
**Contribution:** 2
**Rating:** 2
**Confidence:** 5

**Summary:**

This paper presents a Multi-Scale Hypergraph method that aligns large language models for time series analysis. A hyperedging mechanism is designed to enhance the multi-scale semantic information of the time series semantic space, while a mixture of prompts mechanism is introduced to provide contextual information. Extensive experimental results on real-world datasets demonstrate the superior performance of the proposed method.

**Strengths:**

- The paper introduces a multi-scale hypergraph framework to align LLMs for time series analysis. The proposed hyperedging mechanism and mixture of prompts strategy demonstrate better performance beyond existing single-scale or prompt-based methods.
- This paper is well-written and easy to follow.
- The framework is technically solid and systematically evaluated on 27 real-world datasets covering forecasting, classification, few-shot, and zero-shot learning.

**Weaknesses:**

- The provided code files could not be opened successfully, which raises concerns about the reliability of the reported experimental results.

- The overall framework appears to combine elements from Time-LLM [1] and hypergraph-based multi-scale modeling. As a result, the methodological novelty may be limited unless the authors can clearly articulate the unique contributions beyond this integration.

- The Cross-Modality Alignment (CMA) module used for aligning text and time-series modalities has already been proposed in prior work [2]. Since this paper does not cite that reference, it is unclear whether the presented CMA design introduces any substantive technical advancement.

[1] Time-LLM: Time Series Forecasting by Reprogramming Large Language Models. ICLR 2024.

[2] TimeCMA: Towards LLM-Empowered Multivariate Time Series Forecasting via Cross-Modality Alignment. AAAI 2025, 39(18): 18780–18788.

**Questions:**

- Could the authors clarify how their proposed hyperedging mechanism differs from prior hypergraph-based models such as MSHyper or Ada-MSHyper in terms of functionality?

- Since the CMA module is conceptually similar to that in TimeCMA [2], could the authors explain what improvements are brought by its integration in this work?

- Each file in the code repository link cannot be opened successfully. Could the authors ensure reproducibility by providing a working link or additional implementation details?

- The paper claims novelty in combining multi-scale structures and LLMs. Could the authors discuss whether this combination provides theoretical advantages beyond empirical gains, and how generalizable it is to other LLM backbones?

---

> ### Author Response · Authors · 2025-11-23
>
> **Question 1**. Could the authors clarify how their proposed hyperedging mechanism differs from prior hypergraph-based models such as MSHyper or Ada-MSHyper in terms of functionality?
>
> Thanks for your valuable suggestions and scientific rigor. **In terms of functionality**, the hyperedging mechanism of MSH-LLM differs from prior hyperedge-based methods (e.g., MSHyper and Ada-MSHyper) in the following two key aspects:
>
> 1. Prior hypergraph-based methods (e.g., MSHyper and Ada-MSHyper) are **traditional deep learning-based, task-specific models** that primarily use hypergraphs to model **group-wise interactions between temporal patterns at different scales**. In contrast, MSH-LLM is an **LLM-based, general-purpose** time series analysis model that leverages learnable hyperedges within the hypergraph to **enhance the semantic space of time series**. This approach helps address the semantic sparsity problem of time series data, enabling better alignment with the semantic space of natural language.
>
> 2. Existing hypergraph-based methods (e.g., MSHyper and Ada-MSHyper) **rely on predefined rules or constraint mechanisms** to learn hypergraph structures. In contrast, MSH-LLM learns the hypergraph structures in a **data-driven manner by incorporating learnable parameters and nonlinear transformations**, making it more versatile in capturing implicit interactions and enhancing the multi-scale semantic information of time series.
>
> In addition, we provide a concise summary of **the average performance of these methods across multiple tasks**​ in Table 1. The experimental results demonstrate the effectiveness and generalization capability of MSH-LLM.
>
> Table 1.
>
> | Methods     | Long-Term       | Zero-Shot  | Classification |
> | ----------- | --------------- | ---------- | -------------- |
> | Metric      | MSE MAE         | SMAPE      | Accuracy       |
> | MSHyper     | 0.325 0.344     | /          | /              |
> | Ada-MSHyper | 0.313 0.335     | /          | /              |
> | MSH-LLM     | **0.299 0.325** | **12.719** | **75.38%**     |

---

> ### Author Response · Authors · 2025-11-23
>
> **Question 2**. Since the CMA module is conceptually similar to that in TimeCMA [2], could the authors explain what improvements are brought by its integration in this work?
>
> Thanks for the comments. We need to clarify that **despite the shared nomenclature, the implementation and operational mechanisms** of the cross-modality alignment (CMA) modules in MSH-LLM and TimeCMA **are fundamentally different**. Specifically, the differences can be articulated as follows:
>
> Implementation Details:
>
> The CMA mechanism of TimeCMA adopts a channel-wise similarity retrieval approach. **This method extracts time-series-related representations at a single scale through linear mapping, lacking explicit modeling of complex semantic structures and multi-scale dependencies.** In contrast, MSH-LLM leverages multi-head cross-attention mechanisms to enable multi-scale, cross-modality interactions based on the multi-scale hyperedge features and text prototypes, thereby achieving more precise cross-modality alignment.
>
> Operation Mechanisms:
>
> The CMA mechanism in TimeCMA operates primarily as a **retrieval mechanism**. Its goal is to **query and extract time-series-related representations from a set of predefined, hand-crafted textual prompts**. This process can be seen as a form of feature selection, where the most relevant linguistic cues are retrieved to augment the time series embeddings. In contrast, the CMA module in MSH-LLM operates primarily as an alignment and fusion mechanism. It's designed to align the multi-scale hyperedges features with multi-scale text prototypes generated from the token embeddings of LLMs. This process aims to **align the modality between natural language and time series**.
>
> In addition, we have included TimCMA for comparison. Note that due to its fixed prompt templates and restrictions on LLM selection, we failed to rerun TimeCMA under the unified settings. For a fair comparison, we reran MSH-LLM using the same settings as TimCMA. The experimental results on ETTh1 dataset with the input length T=96 are shown in Table 2, which has also been included in Appendix J.5 of the $\underline{\text{revised paper}}$.
>
> Table 2.
>
> | Methods | MSH-LLM         | TimeCMA         |
> | ------- | --------------- | --------------- |
> | Metric  | MSE MAE         | MSE MAE         |
> | 96      | **0.362** 0.393 | 0.373 **0.391** |
> | 192     | **0.417 0.416** | 0.427 0.421     |
> | 336     | **0.420 0.423** | 0.458 0.448     |
>
> From Table 2, we can observe that MSH-LLM performs better than TimeCMA in almost all cases, **demonstrating the effectiveness of MSH-LLM**.
>
> **Question 3**. Each file in the code repository link cannot be opened successfully. Could the authors ensure reproducibility by providing a working link or additional implementation details?
>
> The anonymous GitHub link has been provided in the Abstract of the $\underline{\text{original paper}}$. However, due to potential issues with the hosting platform (e.g., network latency or server delays), the code may load slowly. **We kindly recommend trying the link again and allowing some time for the content to load fully.** In addition, to facilitate code execution and reproduction, we have provided detailed setup and usage instructions in the README file of the repository. **Please feel free to let us know if you have any further questions about the code repository.**

---

> ### Author Response · Authors · 2025-11-23
>
> **Question 4**. The paper claims novelty in combining multi-scale structures and LLMs. Could the authors discuss whether this combination provides theoretical advantages beyond empirical gains, and how generalizable it is to other LLM backbones?
>
> We sincerely appreciate the valuable feedback from the reviewers and agree that **incorporating theoretical analysis beyond empirical results would enhance the depth and impact of our paper**. In response to the reviewers' suggestions, we have included a theoretical analysis in Appendix K of the $\underline{\text{revised paper}}$. Below, we summarize the key findings:
>
> According to Theorem E.1 in Appendix K, for the self-attention mechanism, if the input token $X_i$ is bounded with mean $\mu$ for $i=1,2,.., N$. Then, under mild conditions, the output token $\hat{{X}}_i$ with high probability converges to $\mu W_i$ at a rate of $\mathcal{O}(n^{-1/2})$, where $W_i$ is the parameter matrix used to compute the value token.
>
> This indicates that the self-attention mechanism used in LLMs can efficiently **converge the output token representations to a stable mean (i.e., the representative semantic center)**. For time series analysis, if there are translation-invariant structures or patterns (e.g., periodicity and trend), the self-attention can help **identify those invariant structures** more effectively by comparing a given token with others. This phenomenon is especially important in few-shot forecasting or high-noise scenarios as it helps avoid overfitting to noise and improves generalization.
>
> However, raw time series data suffer from two main limitations: 1) **Individual time points contain limited semantic information**, making it difficult to reflect structural patterns (e.g., periodicity and trend). 2) **The raw sequence is often corrupted by noise**, resulting in a low signal-to-noise ratio. To address these issues, we introduce **multi-scale hypergraph structures**, which **adaptively connect multiple time points** through learnable hyperedges at different scales. This method can enhance the multi-scale semantic information of time series while reducing irrelevant information interference. **It provides the self-attention mechanism in LLMs with more structured input, enabling self-attention to distinguish between temporal patterns and noise**. As a result, the generalization and robustness of LLMs are improved.
>
> MSH-LLM is designed to enhance the **general ability** of LLMs to understand and process time series data and **can be easily applied to other LLM backbones**. Experimental results on more advanced LLM backbones (e.g., LLaMA-3.18B, Qwen2.5-7B, and DeepSeek-R1-Distill-LLaMA-8B) in Appendix J.1 of the $\underline{\text{original paper}}$ show the generality of MSH-LLM. In addition, to further validate the effectiveness of MSH-LLM, we also evaluate other baseline methods on these advanced LLMs. The experimental results on the ETTh1 dataset with input length T=512 and output length H=96 are presented in Table 3.
>
> Table 3 .
>
> | Methods     | -w L-8B         | -w Q-7B         | -w D-8B         | LLaMA-7B  (Default) |
> | ----------- | --------------- | --------------- | --------------- | ------------------- |
> | Metric      | MSE MAE         | MSE MAE         | MSE MAE         | MSE MAE             |
> | S$^2$IP-LLM | 0.350 0.393     | 0.364 0.395     | 0.362 0.395     | 0.366 0.396         |
> | Time-LLM    | 0.378 0.403     | 0.379 0.413     | 0.378 0.408     | 0.383 0.410         |
> | MSH-LLM     | **0.350 0.377** | **0.352 0.383** | **0.348 0.365** | **0.360 0.388**     |
>
> From Table 3, we can observe that existing LLM4TS methods (i.e., MSH-LLM S$^2$IP-LLM, and Time-LLM) achieve better performance on more advanced LLMs, **demonstrating the significance of the choice of LLM backbones for time series analysis**. In addition, we can observe that MSH-LLM shows a more significant improvement compared to other methods when using more advanced LLM backbones. **This indicates the effectiveness of the framework design, rather than being merely influenced by the LLM backbones.**

---

> ### Author Response · Authors · 2025-11-27
> **Request of Reviewer's attention and feedback**
>
> Dear Reviewer,
>
> Thank you again for your valuable review, which has greatly inspired us to improve our paper further. This is a kind reminder that only a few days remain in the discussion period. We kindly ask if our responses have addressed your concerns.
>
> Following your suggestions, we have strengthened the manuscript by **clarifying the distinctions between MSH-LLM and related hypergraph methods** (e.g., MSHyper and Ada-MSHyper), incorporating **TimeCMA** for comparison, and adding **theoretical analysis** beyond empirical validation.
>
> **Thank you again for your time, dedication, and constructive feedback**. We look forward to hearing your thoughts on our revisions.

---

### Official Review · Reviewer_2He4 · 2025-10-28

**Soundness:** 3
**Presentation:** 3
**Contribution:** 3
**Rating:** 6
**Confidence:** 3

**Summary:**

To address the modality alignment between natural language and time series in LLM-based time series analysis, this paper focuses on multi-scale structures by proposing a novel multi-scale hypergraph method, MSH-LLM. This method integrates multi-scale extraction, hyperedging, cross-modality alignment and mixture of prompts mechanisms to enhance the multi-scale semantic information in LLM-based analysis. Experimental results demonstrate the effectiveness of MSH-LLM.

**Strengths:**

1. This paper introduces a hyperedging mechanism to extract group-wise information from multi-scale temporal features, enhancing multi-scale semantic information of time series semantic space as a novel contribution.

2. This paper designs a cross-modality alignment module to perform multi-scale alignment and obtain richer representations, while the mixture of prompts mechanism enhances the reasoning ability of LLMs.

3. This paper conducts comprehensive experiments on 27 real-world datasets across 5 different applications, demonstrating the effectiveness of MSH-LLM.

**Weaknesses:**

1. In Sec 4.1, the word token embeddings U are transformed into U1 through linear mapping. Is the linear mapping learnable or a semantic-distance-based mapping? Will each text prototype possesses explicit meaning?

2. As the pre-trained LLM is freezed during training according to Figure 1, I am confused whether all the proposed modules will be trained simultaneously. If so, what are the training objectives?

3. Experimental results in Table 1 and 2 show minor improvements between MSH-LLM and the runner-up methods. As the base LLM is the early LLaMa-7B, it is hard to judge whether the performance gap come from the LLM backbones or the method itself.

**Questions:**

See weakness part for details.

---

> ### Author Response · Authors · 2025-11-23
>
> **Question 1**. In Sec 4.1, the word token embeddings U are transformed into U1 through linear mapping. Is the linear mapping learnable or a semantic-distance-based mapping? Will each text prototype possesses explicit meaning?
>
> Thanks for your valuable comments and scientific rigor. **The linear mapping from token embeddings $U$ to $U_1$ is learnable**, parameterized by a trainable weight matrix $W\in \mathbb{R}^{V\times V'}$ and a bias vector $b\in \mathbb{R}^{V'}$. This mapping is data-driven and does not rely on any predefined semantic distance or heuristic rules.
>
> To investigate whether different text prototypes possess explicit semantic meanings, we conduct qualitative analysis by **visualizing the similarity scores between 10 randomly selected text prototypes and word embeddings derived from 3 different word sets**. The visualization results on ETTh1 dataset are given in Figure 8 of the $\underline{\text{revised paper}}$. Below, we provide the main findings:
>
> 1) Prototypes 2, 3, 7, and 8 exhibit strong associations with word set 1 (noun-like time series descriptions), while prototypes 0, 1, and 4 show strong correlations with word set 2 (adjective-like time series descriptions). **This suggests that the prototypes can capture different semantic information, indicating explicit semantic differentiation.**
>
> 2) Although both word set 1 and word set 3 consist of noun-like descriptions, almost all prototypes show weak correlations with word set 3 (name-related words). The reason may be that the text prototypes encode time-series-specific, context-specific semantic information. **The experimental results show that the text prototypes possess explicit meaning.**
>
> **Question 2**. As the pre-trained LLM is freezed during training according to Figure 1, I am confused whether all the proposed modules will be trained simultaneously. If so, what are the training objectives?
>
> As illustrated in Figure 1, **the pre-trained LLMs remain frozen throughout training and all other proposed modules are trainable and are optimized simultaneously in an end-to-end manner.** Since gradient backpropagation follows the chain rule, gradients are propagated layer-by-layer from the loss function to the inputs. Therefore, even for the frozen layers of LLMs, their inputs and outputs remain involved in the gradient flow. However, the gradients within frozen layers are only used for backward computation and **do not lead to updates of the frozen parameters**.
>
> Following existing works [1,2], we adopt MSE as the objective function for long-term time series forecasting and few-shot learning tasks. For short-term time series forecasting and zero-shot learning, we use SMAPE as the objective function. For time series classification, we employ Cross Entropy Loss as the objective function. Thanks to your valuable comments, we have added the clarification in Appendix E of the $\underline{\text{revised paper}}$.
>
> [1] TimesNet: Temporal 2d-variation modeling for general time series analysis. ICLR, 2022.
>
> [2] One fits all: Power general time series analysis by pretrained LM. NeurIPS, 2023.

---

> ### Author Response · Authors · 2025-11-23
>
> **Question 3**. Experimental results in Tables 1 and 2 show minor improvements between MSH-LLM and the runner-up methods. As the base LLM is the early LLaMa-7B, it is hard to judge whether the performance gap come from the LLM backbones or the method itself.
>
> Thanks for your valuable feedback. We would like to respectfully clarify that achieving significant performance improvement across all well-studied datasets is inherently challenging. However, **MSH-LLM still achieves SOTA results in almost all cases**. Specifically, for long-term forecasting, MSH-LLM achieves the **best performance in 40 out of 56 cases**. In addition, on the challenging few-shot learning task (under 5% training data). MSH-LLM gives an average of **4.52% MSE error reduction** compared to the second-best baseline, S$^2$IP-LLM.
>
> To rule out the influence of experimental errors, instead of just showing the MSE and MAE results, we repeat all experiments 3 times and report **the standard deviation and statistical significance level (T-test) of MSH-LLM and the second-best baselines**. The experimental results are shown in Table 1, which has also been included in Appendix J.3 of the $\underline{\text{revised paper}}$.
>
> Table 1.
>
> | Model       | MSH-LLM                       | S$^2$IP-LLM                    | Confidence Interval |
> | ----------- | ----------------------------- | ------------------------------ | ------------------- |
> | Metric      | MSE MAE                       | MSE MAE                        |                     |
> | Weather     | 0.217 ± 0.0006 0.254 ± 0.0005 | 0.223 ± 0.0007 0.259 ± 0.0005  | 99%                 |
> | Electricity | 0.159 ± 0.0006 0.253 ± 0.0003 | 0.163 ± 0.0006 0.258 ± 0.0005  | 99%                 |
> | Traffic     | 0.381 ± 0.0003 0.283 ± 0.0004 | 0.406 ± 0.0003 0.287 ± 0.0004  | 99%                 |
> | ETT (avg.)  | 0.334 ± 0.0020  0.371± 0.0010 | 0.338 ± 0.0014  0.379 ± 0.0010 | 95%                 |
>
> As shown in Table 1, all the statistical significance reaches 95%, indicating that **the performance improvements achieved by MSH-LLM are substantial and consistent across all datasets**.
>
> For a fair comparison, following existing works (e.g., S$^2$IP-LLM and Autotimes), we use LLaMA-7B as the default LLM backbone. However, MSH-LLM is designed to **enhance the general ability of LLMs to understand and process time series data**, rather than being tailored to specific LLMs (e.g., LLaMA-7B). Experimental results on more advanced LLM backbones (e.g., LLaMA-3.18B, Qwen2.5-7B, and DeepSeek-R1-Distill-LLaMA-8B) in Appendix J.2 of the $\underline{\text{original paper}}$ **show the generality of MSH-LLM**. In addition, to further validate the effectiveness of MSH-LLM, we also evaluate other baseline methods on these advanced LLMs. The experimental results on the ETTh1 dataset with input length T=512 and output length H=96 are presented in Table 2.
>
> Table 2.
>
> | Methods | -w L-8B | -w Q-7B | -w D-8B | LLaMA-7B (Default) |
> | --- | --- | --- | --- | --- |
> | Metric | MSE MAE | MSE MAE | MSE MAE | MSE MAE |
> | S$^2$IP-LLM | 0.350 0.393 | 0.364 0.395 | 0.362 0.395 | 0.366 0.396 |
> | Time-LLM | 0.378 0.403 | 0.379 0.413 | 0.378 0.408 | 0.383 0.410 |
> | MSH-LLM | **0.350 0.377** | **0.352 0.383** | **0.348 0.365** | **0.360 0.388** |
>
> From Table 2, we can observe that existing LLM4TS methods (i.e., MSH-LLM S$^2$IP-LLM, and Time-LLM) achieve better performance on more advanced LLMs, demonstrating the significance of the choice of LLM backbones for time series analysis. In addition, we can observe that MSH-LLM shows a more significant improvement compared to other methods when using more advanced LLM backbones. **This indicates the effectiveness of the framework design, rather than being merely influenced by the LLM backbones**.

---

> ### Comment · Reviewer_2He4 · 2025-11-26
> **Thanks for your rebuttal.**
>
> I appreciate that the rebuttal and the newly revised manuscript address most of my concerns in model details and experimental analysis. However, I am still not confident whether this method is effective by switching to another advanced LLM backbone (e.g., Qwen2.5, Deepseek, GLM, even a smaller-sized LLM is ok), as this is a critical point reflecting the adaptability of the proposed method. **On this basis, considering the authors' efforts and the quality of the manuscript, I am willing to update my rate to 7 (if there is such a choice) but not enough for 8.** I can only keep my score at this moment.

---

> > ### Author Response · Authors · 2025-11-26
> >
> > Thank you for your comments.
> > Actually, we did conduct the variant experiments as you suggested, but we accidentally forgot to include the corresponding results when copying the rebuttal content. We have now added the experimental results after Question 3 (Q3)​ in our rebuttal, and they are also included in the revised manuscript. We sincerely apologize for any confusion caused by this oversight. We appreciate your understanding.

---

> > > ### Comment · Reviewer_2He4 · 2025-11-26
> > > **Thanks for your update!**
> > >
> > > I have reviewed your update about switching the LLM backbones, and the consistent improvement addresses my concern about the adaptability. On this basis, I will raise my score to 8.

---

> > > > ### Author Response · Authors · 2025-11-26
> > > >
> > > > We are grateful that our responses have addressed your concerns. We truly appreciate the time you took to review our paper in detail and for sharing such valuable feedback with us.

---

### Official Review · Reviewer_iwbv · 2025-10-30

**Soundness:** 3
**Presentation:** 2
**Contribution:** 2
**Rating:** 4
**Confidence:** 4

**Summary:**

The paper addresses the problem of multi-scale semantic misalignment between large language models (LLMs) and time series data by proposing MSH-LLM, a cross-modal alignment framework based on multi-scale hypergraphs. The core innovation lies in the hyperedge mechanism, which enhances the multi-scale semantic representation of time series, together with a hybrid prompting strategy that activates the temporal reasoning capability.

**Strengths:**

1. The paper insightfully identifies the multi-scale structural misalignment between natural language and time as a fundamental bottleneck for applying LLMs to time-series analysis.

2. By introducing learnable hyperedges and a sparsification strategy, the proposed method not only mitigates the noise sensitivity of traditional patch-based approaches but also captures implicit group-level interactions within time series data.

**Weaknesses:**

1. The paper does not justify the node–hyperedge similarity in the hyperedging mechanism (Eqs. 2–3), nor does it analyze the theoretical relation between the number of hyperedges M^s and the time-series dimensionality D.

2. The work does not report inference latency on ETTh1 with H=720, so it is unclear whether the method is suitable for real-time scenarios such as power-load dispatch. Memory consumption is also missing; GPU usage on high-dimensional data is not reported, limiting an assessment of deployability.

3. Within the capability-enhancing prompts, “logical reasoning appears to overlap with 'time-series reasoning, but there is no ablation to test redundancy.

**Questions:**

1. Please include training-time trajectories of the hyperedge embeddings E_hyper^s. An analysis that maps dominant embedding directions to known multi-scale patterns in the data, for example, daily and weekly cycles in ETT, could substantially improve the interpretability of the hyperedging mechanism.

2.  Please evaluate the robustness of MSH-LLM under extreme conditions. Specifically, include results on ETTh1 subsets with 10% to 20% injected anomalies, ultra-long forecasting with H=1008, and Electricity subsets with 5% missing data. A comprehensive discussion of how the model behaves across these challenging scenarios—and particularly where performance begins to degrade—would provide valuable insight into the robustness boundaries of MSH-LLM.

3.  Please provide inference latency and memory usage on an NVIDIA A100. For example, report single-pass inference time on ETTh1 with H=720and the peak GPU memory under batch size =32. These measurements are important to assess real-time feasibility.

4.  Please run targeted ablations that remove only the “logical reasoning” prompt or only the “time-series reasoning” prompt, and report the resulting performance changes. This would help determine whether the two components are redundant and if they can be merged to simplify the prompt design.

---

> ### Author Response · Authors · 2025-11-23
>
> **Question 1**. Please include training-time trajectories of the hyperedge embeddings E_hyper^s. An analysis that maps dominant embedding directions to known multi-scale patterns in the data, for example, daily and weekly cycles in ETT, could substantially improve the interpretability of the hyperedging mechanism.
>
> Thanks for your insightful comments and scientific rigor. We perform qualitative analysis to investigate **the training-time trajectories of the hyperedge embeddings**. The t-SNE visualization of hyperedge embeddings is given in Figure 6 of the $\underline{\text{revised paper}}$. From Figure 6, we can discern the following tendencies:
>
> 1). As training progresses, hyperedge embeddings at different scales form distinct clusters. This indicates that MSH-LLM **is able to distinguish and capture multi-scale temporal patterns**. In addition, even within the same scale, different hyperedge embeddings reside in distinct clusters, indicating the ability of **MSH-LLM in capturing diverse temporal patterns within the same scale**.
>
> 2). From Figure 6(a) to Figure 6(c), we can observe that embeddings of large-scale hyperedges **form distinct clusters earlier during training**, while embeddings of small-scale hyperedges **gradually separate from the large-scale** clusters over time. This suggests that during the early stages of training, the model is more focused on capturing **coarse-grained temporal patterns (e.g., weekly patterns)**, and later shifts its focus to learning **finer-grained temporal patterns (e.g., hourly and daily patterns)**.
>
> **Question 2**. Please evaluate the robustness of MSH-LLM under extreme conditions. Specifically, include results on ETTh1 subsets with 10% to 20% injected anomalies, ultra-long forecasting with H=1008, and Electricity subsets with 5% missing data. A comprehensive discussion of how the model behaves across these challenging scenarios—and particularly where performance begins to degrade—would provide valuable insight into the robustness boundaries of MSH-LLM.
>
> To evaluate the robustness of the proposed method, we compare MSH-LLM with baselines (i.e., S$^2$IP-LLM, Time-LLM, and FPT) across three challenging scenarios: **forecasting with anomaly injection, ultra-long forecasting, and forecasting with missing data**. The corresponding results are presented below, which have also been included in Appendix J.3 of the $\underline{\text{revised paper}}$. Note that to quantify robustness, we compute the performance drop rate (PDR)  as:
>
> $\text{PDR} = \frac{\Gamma - \hat{\Gamma}}{\Gamma}$
>
> where $\Gamma$ and $\hat\Gamma$ are forecasting results and forecasting results under challenging scenarios, respectively. Higher PDR values indicate lower robustness. The reported PDR is averaged across the MSE and MAE metrics to provide a comprehensive evaluation.
>
> **Forecasting With Anomaly Injection.** We conduct experiments by injecting randomly generated anomalies in the training data. The anomaly rate varies from 10% to 20%. The experiments are conducted on ETTh1 dataset with the input length set to 512 and output length set to 96. Table 1 summarizes the results of forecasting with anomaly injection.
>
> Table 1.
>
> | Methods | MSH-LLM                | S$^2$IP-LLM        | Time-LLM           | FPT                 |
> | ------- | ---------------------- | ------------------ | ------------------ | ------------------- |
> | Metric  | MSE MAE PDR            | MSE MAE PDR        | MSE MAE PDR        | MSE MAE PDR         |
> | 0%      | **0.360 0.388   /**    | 0.366 0.396   /    | 0.383 0.410   /    | 0.379 0.402   /     |
> | 10%     | **0.374 0.393 2.589**  | 0.712 0.574 69.743 | 0.398 0.419 3.056  | 0.410 0.393 2.970   |
> | 15%     | **0.425 0.427 14.053** | 0.723 0.578 71.750 | 0.443 0.435 10.812 | 0.741 1.103 134.946 |
> | 20%     | **0.732 0.582 76.667** | 0.773 0.598 81.106 | 0.751 0.589 69.871 | 0.935 1.421 200.092 |
>
> From Table 1, we can obtain the following tendencies: 1) MSH-LLM achieves the best performance in almost all cases, showing its superior ability **in time series forecasting even under scenarios with anomaly injection**. 2) Although the performance of all methods declines as the anomaly ratio increases, MSH-LLM exhibits a slower performance degradation compared to the other methods, demonstrating **its robustness for forecasting with anomaly injection**. 3) When the anomaly ratio reaches about 20%, the PDR value of MSH-LLM is greater than 20%, indicating that the robustness boundary of MSH-LLM is **near 20% anomaly injection**.

---

> ### Author Response · Authors · 2025-11-23
>
> **Ultra-Long-Term Forecasting.** We conduct ultra-long-term time series forecasting by taking a fixed input length (T=512) to predict ultra-long horizons (H={1008, 1440, 1800}). Table 2 summarizes the results of ultra-long-term time series forecasting.
>
> Table 2.
>
> | Methods | MSH-LLM         | S$^2$IP-LLM | Time-LLM    | FPT         |
> | ------- | --------------- | ----------- | ----------- | ----------- |
> | Metric  | MSE MAE         | MSE MAE     | MSE MAE     | MSE MAE     |
> | 1008    | **0.463 0.498** | 0.543 0.520 | 0.478 0.475 | 0.527 0.576 |
> | 1440    | **0.516 0.513** | 0.806 0.642 | 0.547 0.521 | 0.594 0.716 |
> | 1800    | **0.648 0.557** | 0.940 0.725 | 0.683 0.587 | 0.660 0.886 |
>
> From Table 2, we can observe that MSH-LLM achieves SOTA results on almost all cases, showing the effectiveness of MSH-LLM for ultra-long-term time series forecasting. In addition, although all baselines suffer from performance drops when increasing forecasting horizons, MSH-LLM **declines more gradually**. The reason may be that the multi-scale hypergraph structure **enhances the ability of LLMs in understanding and processing ultra-long-term time series**.
>
> **Forecasting With Missing Data.** We conduct forecasting with missing data by randomly masking the training data. The experiments are conducted on Electricity dataset with the input length set to 512 and output length set to 96. Table 3 summarizes the results of forecasting with missing data.
>
> Table 3.
>
> | Methods | MSH-LLM                | S$^2$IP-LLM        | Time-LLM           | FPT                |
> | ------- | ---------------------- | ------------------ | ------------------ | ------------------ |
> | Metric  | MSE MAE PDR            | MSE MAE PDR        | MSE MAE PDR        | MSE MAE PDR        |
> | 0%      | **0.360 0.388**   /    | 0.366 0.396   /    | 0.383 0.410   /    | 0.379 0.402   /    |
> | 5%      | **0.368 0.393 3.511**  | 0.385 0.403 3.479  | 0.392 0.416 1.907  | 0.392 0.431 5.307  |
> | 10%     | **0.409 0.421 11.058** | 0.432 0.447 15.456 | 0.451 0.449 13.633 | 0.478 0.483 22.704 |
>
> From Table 3, we can obtain the following tendencies: 1) Existing LLM-based methods show little performance degradation with 5% missing data. The reason may be that LLM4TS methods can **leverage transferable knowledge learned from large-scale corpora of sequence**s, thereby enhancing their abilities in understanding and reasoning time series. 2) MSH-LLM performs better than other LLM4TS methods, the reason is that **the hyperedging mechanism can capture group-wise interactions**, which increase the robustness of LLM in forecasting with missing data. 3) When the missing data ratio reaches about 10%, the PDR value of MSH-LLM is greater than 10%, indicating that **the robustness boundary of MSH-LLM is near 10% missing data**.
>
> **Question 3**. Please provide inference latency and memory usage on an NVIDIA A100. For example, report single-pass inference time on ETTh1 with H=720 and the peak GPU memory under batch size =32. These measurements are important to assess real-time feasibility.
>
> We greatly appreciate your scientific rigor regarding the computation cost. Following your suggestions, we conducted additional experiments on ETTh1 dataset with the input length of 512 and output length of 720 using a batch size of 32. The experimental results, including inference time, number of parameters, GPU occupations, and MAE results, are given in Table 4, which has also been included in Appendix J.4 of the $\underline{\text{revised paper}}$.
>
> Table 4.
>
> | Methods     | Inference Time | #Parameters    | GPU Occupation | MAE Results |
> | ----------- | -------------- | -------------- | -------------- | ----------- |
> | MSH-LLM     | 0.104s         | 75,852,238     | 7,872MB        | **0.451**   |
> | S$^2$IP-LLM | 0.442s         | 63,636,512     | 9,991MB        | 0.459       |
> | Time-LLM    | 0.116s         | 53,441,968     | 5,403MB        | 0.460       |
> | FPT         | **0.015**s     | **36,209,616** | **2,623**MB    | 0.463       |
>
> From Table 4, we can observe that FPT has the fewest parameters and runs faster than other LLM4TS methods, but it gets the worst forecasting results. Compared with  S$^2$IP-LLM and Time-LLM, **although MSH-LLM has a larger number of parameters, it runs fastest due to the matrix sparsity strategy in the model and the optimization of hypergraph computation provided by torch geometry**. Overall, considering both the forecasting performance improvement and the computation cost, MSH-LLM demonstrates its superiority over existing methods.

---

> ### Author Response · Authors · 2025-11-23
>
> **Question 4**. Please run targeted ablations that remove only the “logical reasoning” prompt or only the “time-series reasoning” prompt, and report the resulting performance changes. This would help determine whether the two components are redundant and if they can be merged to simplify the prompt design.
>
> To investigate the impact of the logical thinking prompt and time series reasoning correlated prompt, we conduct ablation studies by carefully designing the following variants:
>
> -w/o LR: It removes the logical thinking prompt used in the MoP mechanism.
>
> -w/o TSR: It removes the time series reasoning correlated prompt used in the MoP mechanism.
>
>    Table 5.
>
> | Variation | -w/o LR     | -w/o   TSR  | MSH-LLM         |
> | --------- | ----------- | ----------- | --------------- |
> | Metric    | MSE MAE     | MSE MAE     | MSE MAE         |
> | 96        | 0.372 0.274 | 0.370 0.268 | **0.365 0.270** |
> | 192       | 0.389 0.287 | 0.383 0.283 | **0.372 0.281** |
> | 336       | 0.390 0.282 | 0.379 0.280 | **0.385 0.279** |
>
>  The experimental results on Traffic dataset are shown in Table 5, which has also been included in Appendix H of the $\underline{\text{revised paper}}$. From Table 5, we can observe that -w/o LR performs better than -w/o TSR, indicating that **the logical reasoning prompt plays a more critical role than the time series reasoning correlated prompt**. In addition, -w/o LR and -w/o TSR perform worse than MSH-LLM, **showing the effectiveness of the logical reasoning prompts and time series reasoning correlated prompts**, respectively.

---

> ### Author Response · Authors · 2025-11-27
> **Request of Reviewer's attention and feedback**
>
> Dear Reviewer,
>
> Thank you again for your valuable review, which has greatly inspired us to improve our paper further. This is a kind reminder that only a few days remain in the discussion period. We kindly ask if our responses have addressed your concerns.
>
> The paper has been strengthened in line with your suggestions. Key enhancements include **a qualitative analysis of hyperedge embeddings**, **robustness evaluations** of MSH-LLM under extreme conditions, and **additional ablation studies** focusing on the MoP mechanism.
>
> **Thank you again for your time, dedication, and constructive feedback**. We look forward to hearing your thoughts on our revisions.

---

### Author Response · Authors · 2025-11-24
**Summary of Revisions**

We sincerely thank all the reviewers for their insightful reviews and valuable comments, which are instructive for us to improve our paper further.

Overall, reviewers acknowledge that **MSH-LLM first aligns time-text features via cross-view attention**, by introducing the hyperedging mechanism and sparfication strategy, the model can capture group-wise interactions within time series data, and the mixture of prompts mechanism enhances the reasoning ability of LLMs (Reviewers iwbv, 2He4, GvBV, and GFpP). They highlight that the paper is **well-written and easy to follow**, the framework is **technically solid**, and the exposition is **logically coherent** (Reviewers GvBV and GFpP). Empirically, reviewers deem MSH-LLM is **novel, rigorous, insightful, and systematically evaluated** (Reviewer iwBV, 2He4, and GVBV).

The reviewers also raise insightful and constructive concerns. We made every effort to address all of them by providing the requested results and additional evidence. Below is a summary of the major revisions:

**Add more ablation studies and model analysis (Reviewers iwbv, 2He4, GvBV, and GFpP)**: Following the suggestions of the reviewers, we have added more than 15 ablation studies to investigate the effectiveness of the hyperedging mechanism, the mixture of prompts (MoP) mechanism, the cross-modality alignment (CMA) module, and the LLM backbones. In addition, we have added more robustness analysis and computation cost analysis to investigate the performance of MSH-LLM across different challenging scenarios.

**Add additional baselines and datasets (Reviewers GvBV and GFpP)**: Following the suggestions of the reviewers, we have added 5 anomaly detection datasets for comparison and conducted comparisons with an LLM-based time series forecasting model (i.e., TimeCMA) on ETTh1 datasets. The experimental results demonstrate the effectiveness of MSH-LLM over existing methods.

**Provide more qualitative analysis and theoretical proof (Reviewers iwbv, 2He4, and GvBV)**: Following the suggestions of the reviewers, we have added qualitative analysis to investigate the training-time trajectories of the hyperedging embeddings, the effectiveness of text prototypes. In addition, we have also provided theoretical proof to investigate why the multi-scale hypergraph structures are useful for LLMs in understanding temporal patterns.

**After 7 full days of experiments (with 4 A100-80 GPUs), we have added more than 100 new experimental results to address the mentioned issues. All the revisions have been included in the revised paper**. We sincerely appreciate the valuable suggestions from the reviewers, which have been instrumental in improving the quality of our paper. We would be happy to address any further questions.

If all concerns have been sufficiently addressed, we kindly request that you consider raising the overall score. We truly appreciate the time and effort you have dedicated to reviewing our work, and we are grateful for your continued recognition and support.

---

### Author Response · Authors · 2025-12-03
**Summary of Rebuttal and Discussion Phase for Paper 12086**

Dear Area Chairs, Senior Area Chairs, and Program Chairs,

Due to the OpenReview security incident, the rebuttal-discussion phase was cut short and only one reviewer responded to our rebuttal. To assist your assessment under these circumstances, we briefly summarize how the main concerns have been addressed. **All corresponding revisions are included in the updated manuscript and highlighted in $\textcolor{blue}{\text{blue}}$**.

**1. Consensus on strengths**

All four reviewers express a consistently positive view on MSH-LLM: They recognize it as the **first multi-scale alignment work** for time series analysis, noting that by introducing the hyperedging mechanism and sparsification strategy, the model can capture group-wise interactions within time series data and that the mixture of prompts mechanism **enhances the reasoning ability of LLMs**. They also highlight that the paper is **well-written** and **easy to follow**, that the framework is **technically solid**, and that the exposition is **logically coherent**. They also acknowledge that MSH-LLM achieves consistent state-of-the-art performance on **27 real-world datasets** across 5 different applications.

**2. How key concerns were resolved**

**A. Complexity, scalability, and robustness**

To further evaluate the performance of MSH-LLM under extreme conditions, we compare it with baselines across three challenging scenarios: forecasting with anomaly injection, ultra-long forecasting, and forecasting with missing data. The substantial and consistent performance improvement demonstrates the effectiveness of MSH-LLM over existing methods. In addition, the additional computational cost on ETTh1 dataset demonstrates that MSH-LLM maintains both efficiency and scalability in time series data processing.

**B. Baselines, datasets, and ablation studies**

Following the suggestions of the reviewers, we included TimeCMA for comparison. The experimental results demonstrate the effectiveness of MSH-LLM. In addition, we added 5 anomaly detection datasets (i.e., SMD, MSL, SMAP, SWaT, and PSM) for comparison. The experimental results further demonstrate the generality of MSH-LLM. We clarified that prior hypergraph-based methods (e.g., MSHyper and Ada-MSHyper) are traditional deep learning-based, task-specific models that primarily rely on predefined rules or constraint mechanisms to learn hypergraph structures. In contrast, MSH-LLM is an LLM-based, general-purpose time series analysis model that learns the hypergraph structures in a data-driven manner by incorporating learnable parameters and nonlinear transformations. Finally, more ablation studies demonstrate the effectiveness of the hyperedging mechanism, the mixture of prompts (MoP) mechanism, and the cross-modality alignment (CMA) module.

**C. Qualitative analysis and theoretical analysis**

Following the reviewers' suggestions, we added qualitative analysis to investigate the training-time trajectories of the hyperedging embeddings and the effectiveness of text prototypes. The t-SNE visualization of hyperedge embeddings suggests that during the early stages of training, the model is more focused on capturing coarse-grained temporal patterns (e.g., weekly patterns), and later shifts its focus to learning finer-grained temporal patterns (e.g., hourly and daily patterns). In addition, the visualization results show that the text prototypes can encode time-series-specific, context-specific semantic information. Finally, the theoretical analysis provides that the hyperedging mechanism provides self-attention in LLMs with more structured input, enabling self-attention to distinguish between temporal patterns and noise. Experimental results show that MSH-LLM can be easily applied to other LLM backbones.

**3.Conclusion**

After the rebuttal, Reviewer 2He4 acknowledged that our response **addressed his concerns, championed our paper, and $\textcolor{red}{\text{raised the score to }8}$**. Notably, the two reviewers who gave lower scores focused primarily on the robustness of MSH-LLM, the completeness of baselines, and clarity, rather than on fundamental flaws in the core multi-scale alignment paradigm. In the rebuttal, we targeted these points by incorporating **substantial new experimental evidence** and **clarifying our assumptions and positioning**. In addition, Reviewer GFpP has also **championed our paper**, acknowledging that “The exposition is well-structured and logically coherent, with rigorous reasoning throughout.” Taken together with the reviewers’ consensus on the novelty and empirical strength of MSH-LLM, the additional analyses now incorporated in the revised manuscript lead us to believe that **the paper adequately addresses the main concerns, makes a solid contribution by combining multi-scale alignment with the MoP mechanism in time series analysis, and is well positioned to meet the standard for acceptance**.

We sincerely thank the AC for considering our work under these unusual circumstances.

---

### Meta-Review · Area_Chair_ncep · 2026-01-06

**Summary:**

This paper proposes a multi-scale hypergraph-based framework for aligning large language models with time series data.

The reviewers generally agree on the following strengths:
- The paper addresses an important problem, namely the semantic misalignment between natural language and temporal data, which is a key obstacle in applying LLMs to time series analysis.
- The core idea of modeling multi-scale temporal structure via hypergraphs is novel and compelling, and the overall method design is technically sound.
- The empirical evaluation is extensive, and the reported performance is consistently strong across tasks and datasets.
- The paper is generally well written and easy to follow.

The reviewers also raised several concerns; however, these primarily relate to clarification and additional experimental analysis, which have been adequately addressed by the authors.

Overall, the paper makes a solid contribution, and thus I recommend acceptance.

**Reviewer Concerns:**

The concerns were mostly related to clarification, including comparisons with related work, and additional experiments on ablation and scalability. These issues have been adequately addressed by the authors.

**Reviewer Scores:**

As discussed above, the reviewers’ concerns could be easily addressed during the rebuttal phase. I believe that most reviewers increased their scores at least slightly following the rebuttal. Reviewer 2He4 explicitly indicated a willingness to raise the score; however, this was not a decisive factor in my recommendation.

---

### Decision · Program_Chairs · 2026-01-26

Accept (Poster)